# Dual-Emitting Molecularly Imprinted Nanopolymers for the Detection of CA19-9

**DOI:** 10.3390/biomedicines13071629

**Published:** 2025-07-03

**Authors:** Eduarda Rodrigues, Ana Xu, Rafael C. Castro, David S. M. Ribeiro, João L. M. Santos, Ana Margarida L. Piloto

**Affiliations:** 1CIETI-LabRISE, ISEP, Polytechnic of Porto, Rua Dr. António Bernardino de Almeida 431, 4249-015 Porto, Portugal; edmfr@isep.ipp.pt (E.R.); naaxu@isep.ipp.pt (A.X.); 2LAQV, REQUIMTE, Laboratory of Applied Chemistry, Department of Chemical Sciences, Faculty of Pharmacy, University of Porto, Rua de Jorge Viterbo Ferreira nº 228, 4050-313 Porto, Portugal; rafael.castro.cl@hotmail.com (R.C.C.); dsmribeiro@gmail.com (D.S.M.R.); joaolms@ff.up.pt (J.L.M.S.)

**Keywords:** dual-emitting molecularly imprinted nanopolymers (dual@nanoMIPs), yellow-emitting quantum dots (y-QDs), blue-emitting carbon dots (b-CDs), carbohydrate antigen 19-9 (CA19-9), pancreatic cancer (PC), optical sensor

## Abstract

**Background/Objectives:** Carbohydrate antigen 19-9 (CA19-9) is a clinically established biomarker primarily used for monitoring disease progression and recurrence in pancreatic and gastrointestinal cancers. Accurate and continuous quantification of CA19-9 in patient samples is critical for effective clinical management. This study aimed to develop dual-emitting molecularly imprinted nanopolymers (dual@nanoMIPs) for ratiometric and reliable detection of CA19-9 in serum. **Methods:** Dual-emitting nanoMIPs were synthesized via a one-step molecular imprinting process, incorporating both blue-emitting carbon dots (b-CDs) as internal reference fluorophores and yellow-emitting quantum dots (y-QDs) as responsive probes. The CA19-9 template was embedded into the polymer matrix to create specific recognition sites. Fluorescence measurements were carried out under 365 nm excitation in 1% human serum diluted in phosphate-buffered saline (PBS). **Results**: The dual@nanoMIPs exhibited a ratiometric fluorescence response upon CA19-9 binding, characterized by the emission quenching of the y-QDs at 575 nm, while the b-CDs emission remained stable at 467 nm. The fluorescence shift observed in the RGB coordinates from yellow to green in the concentration range of CA19-9 tested, improved quantification accuracy by compensating for matrix effects in serum. A linear detection range was achieved from 4.98 × 10^−3^ to 8.39 × 10^2^ U mL^−1^ in serum samples, with high specificity and reproducibility. **Conclusions:** The dual@nanoMIPs developed in this work enable a stable, sensitive, and specific detection of CA19-9 in minimally processed serum, offering a promising tool for longitudinal monitoring of cancer patients. Its ratiometric fluorescence design enhances reliability, supporting clinical decision-making in the follow-up of pancreatic cancer.

## 1. Introduction

The rapid and reliable detection of biological targets, such as DNA, antibodies, and proteins, is critical in modern biomedical research and clinical diagnostics, enabling timely assessment of health conditions and disease progression [1,2,3,4,5,6]. Among these biomarkers, carbohydrate antigen 19-9 (CA19-9) is particularly important due to its association with pancreatic cancer, a disease with high mortality rates primarily due to late-stage diagnosis. Although CA19-9 is not exclusively specific to pancreatic cancer, it is routinely used in clinical practice for early diagnosis and risk assessment, complementing imaging and other diagnostic modalities [7,8,9]. Traditional detection methods for CA19-9, including enzyme-linked immunosorbent assays (ELISAs) [10,11], surface-enhanced Raman scattering (SERS) [12], fluorescence-based assays [13,14], electrochemical sensing [15,16,17,18], and mass spectrometry [19,20], offer high sensitivity and specificity. However, these techniques typically require sophisticated instrumentation, complex sample preparation, and trained personnel, factors that limit their application in rapid screening and point-of-care (PoC) settings. In response, point-of-care tests (PoCTs) have gained attention for their potential to provide accessible, rapid, and user-friendly diagnostic solutions [21,22,23,24]. A central challenge in developing effective PoCTs lies in achieving high sensitivity and specificity using stable, low-cost biorecognition elements. Molecularly imprinted polymers (MIPs) have emerged as promising synthetic alternatives to antibodies in sensor design due to their excellent chemical stability, reusability, and tunable selectivity [25,26]. Despite these advantages, conventional MIP-based fluorescent sensors can suffer from signal instability and environmental interference, particularly when applied in complex biological matrices [27,28,29]. To address these limitations, this study presents dual-emitting molecularly imprinted nanopolymers (dual@nanoMIPs) for the sensitive and selective detection of CA19-9. To the best of our knowledge, this is the first report integrating both blue-emitting carbon dots (b-CDs) and yellow-emitting quantum dots (y-QDs) into a MIP framework for ratiometric fluorescence sensing of CA19-9 [30]. The dual-emission design enables internal referencing, enhancing signal reliability by compensating for environmental fluctuations, such as pH changes, temperature variations, and photobleaching [31,32,33,34]. The rationale behind this approach is twofold: First, the molecular specificity imparted by MIPs ensures targeted binding of CA19-9, while the integration of fluorescent nanodots enhances signal generation and readout accuracy [35,36,37,38,39]. Second, the use of ratiometric fluorescence—based on the intensity ratio between y-QDs (signal channel) and b-CDs (reference channel)—mitigates external interference and improves reproducibility compared to single-emission systems. While traditional QD-based immunoassays offer high brightness and tunable optical properties [40,41,42], their reliance on fragile and expensive biological recognition elements limits scalability and robustness. In contrast, MIP-based platforms offer a synthetic, cost-effective alternative, although they typically face challenges, such as low imprinting efficiency or signal interference [25,26,43]. By combining the optical advantages of QDs with the robustness of MIPs and the stability of internal referencing via b-CDs, the proposed dual@nanoMIPs sensor overcomes these drawbacks. The resulting sensor exhibits high selectivity, minimal cross-reactivity, and a wide dynamic detection range (4.98 × 10^−3^ to 8.39 × 10^2^ U mL^−1^) in 1% human serum in PBS. This system represents a significant step forward in the development of next-generation PoC diagnostic tools for early pancreatic cancer screening, recurrence monitoring, and clinical decision-making.

## 2. Materials and Methods

### 2.1. Materials

Tellurium powder (200 mesh, 99,8%), sodium borohydride (NaBH_4_, 99%), cadmium chloride hemi(pentahydrate) (CdCl_2_.2,5H_2_O, 99%), sodium hydrogen carbonate (NaHCO_3_), and sodium carbonate decahydrate (Na_2_CO_3_.10H_2_O) were purchased from Sigma-Aldrich. Unconjugated human carbohydrate antigen (CA19-9) was purchased from Biorbyt Ltd., Cambridge, UK. Absolute ethanol (>99%) was obtained from Riedel-de Haën, Seelze, Germany, and acrylamide (AAM), bisacrylamide (MBA), 2-aminoethyl methacrylate hydrochloride (AEMH), 3-mercaptopropionic acid (MPA, 99%), and tetramethyl ethylenediamine (TEMED) were obtained from TCI. Phosphate-buffered saline (PBS) tablets were obtained from Amresco, ammonium persulfate (APS) was obtained from Analar Normapur, and human normal (HN) serum was purchased from PZ CORMAY S.A., Łomianki, Poland, and reconstituted according to the manufacturer’s instructions. CA19-9 recombinant antigen produced in *E. coli* with a molecular weight of approximately 210 kDa was acquired from Biorbyt, human serum albumin (HSA) and creatinine (CREA) were purchased by Sigma, and matrix metalloproteinase 7 (MMP7) and carcinoembryonic antigen (CEA) were purchased from abbexa. All fluorescence measurements were performed with a QS high-precision cell with a 1 mm light path made of quartz suprasil^®^ from Hëllma^®^Analytics. All solutions were prepared with water from a Milli-Q system (specific conductivity < 0.1 μS cm^−1^) and chemicals of analytical reagent grade quality. Reagents were not subject to further purification. Information related to instrumentation and optical measurements can be found in the Appendix A.

### 2.2. Methods

UV-vis absorption spectra were recorded over 430–730 nm using a Thermo Scientific Evolution 220 spectrophotometer (Thermo Fisher Scientific Inc., Waltham, MA, USA). 

FTIR spectra were collected on a Nicolet iS10 spectrometer (Thermo Fisher Scientific Inc., Waltham, MA, USA) with an ATR accessory (diamond crystal) at 16 cm^−1^ resolution over 800–4000 cm^−1^.

Scanning electron microscopy (SEM) and energy-dispersive X-ray spectroscopy (EDS) analyses were performed on a FEI Quanta 400 FEG at 15 kV (FEI Company, Hillsboro, OR, USA).

Fluorescence spectra were recorded on a Lumina spectrometer (Thermo Fisher Scientific Inc., Waltham, MA, USA), with a 150 W xenon-arc lamp. Unless otherwise noted, samples were excited at 380 nm with emission recorded between 430 and 730 nm. Both excitation and emission slits were set to 20 nm, and measurements were conducted in 1 mm quartz cells.

### 2.3. Synthesis of Blue-Emitting Carbon Dots (b-CDs)

b-CDs were synthesized following a modified hydrothermal method [39,40]. Citric acid (1.1 g) and ethylenediamine (0.7 mL) were dissolved in water to form a 10% (*w*/*v*) solution. The pH was adjusted to 4.0 using 1 M HCl. The solution was sealed in a 50 mL Teflon-lined autoclave and heated at 260 °C for 4 h (Figure 1A). After cooling to room temperature (~22 °C), the product was purified by dialysis (1000 Da cutoff) against Milli-Q water for five days. The purified b-CDs solution was stored in the dark at rt and diluted in PBS to an absorbance of 0.2 at 320 nm prior to use.

### 2.4. Synthesis of Yellow-Emitting CdTe Quantum Dots (y-QDs)

y-QDs were prepared based on Zou et al. with slight modifications [41]. CdCl_2_ (4.2 × 10^−3^ mol) and MPA (7.2 × 10^−3^ mol) were dissolved in 100 mL of ultrapure water, and the pH was adjusted to 11.5 with 1 M NaOH.

Separately, a Te^2−^ precursor was generated by reacting tellurium powder (2.9 × 10^−3^ mol) with NaBH_4_ (4.8 × 10^−3^ mol) in 3 mL of degassed water at 100 °C under N_2_ purging for 30 min. The Te^2−^ solution was rapidly injected into the Cd^2+^/MPA solution, maintaining a molar ratio of Cd^2+^:Te^2−^:MPA of 1:0.1:1.7, and the mixture was refluxed for 4 h (Figure 1B). The colloidal solution was precipitated with ethanol and centrifuged at 4000 rpm for 5 min at 22 °C. The collected solid was dried under dark conditions and stored away from light until use.

### 2.5. Preparation of Dual-Emitting Nanodots (Dual@Nanodots)

A suspension of y-QDs (1 mg mL^−1^ in PBS 10 mM, pH 7.4) was mixed with 10 µL of the b-CDs suspension (Abs at 320 nm = 0.1), and the final volume was adjusted to 1 mL with PBS. This mixture served as the dual-emission fluorescent probe for further polymer imprinting.

### 2.6. Synthesis of dual@nanoMIPs

Dual@nanoMIPs were synthesized via a free-radical polymerization method under nitrogen to minimize oxidative inhibition (Figure 1C). All aqueous solutions were deoxygenated by N_2_ purging prior to use. To form the pre-polymerization complex, CA19-9 at 1 kU mL^−1^ and AEMH at 33.7 mg mL^−1^ were incubated in PBS; 10 mM, pH 7.4 for 20 min at rt, facilitating non-covalent interactions between the template and functional monomer. Subsequently, y-QDs at 1 mg mL^−1^ were added, and the volume was adjusted to 1 mL. The mixture was centrifuged (4000 rpm, 2 min, 22 °C), and the supernatant was discarded to concentrate the complex. The resulting pellet was resuspended in PBS containing b-CDs (10 µL, diluted in PBS to a maximum absorption of 0.1 at 320 nm), along with AAM at 44.1 mg mL^−1^, MBA at 22.3 mg mL^−1^, APS at 13.6 mg mL^−1^ and TEMED at 13.3 mg mL^−1^. The final volume was readjusted to 1 mL, and polymerization was carried out at rt for 30 min. After polymerization, the nanoparticles were collected by centrifugation (4000 rpm, 2 min, 22 °C), and the template was removed using carbonate/bicarbonate buffer (10 mM, pH 9.8). The pellets were washed in three cycles of resuspension and centrifugation until UV-vis analysis confirmed the absence of free CA19-9 in the supernatant of the dual@nanoMIPs. For comparison, a second batch of dual@nanoMIPs was synthesized using a higher CA19-9 concentration (10 kU mL^−1^) to evaluate imprinting efficiency and selectivity. Control non-imprinted polymers (dual@nanoNIPs) were prepared using the same protocol but without the template CA19-9 during the pre-complexation step. Finally, the dual@nanoMIPs were stored as follows: for long-term storage, they were dried and kept protected from light at rt after a water wash and centrifugation; for short-term use (≤15 days), they were stored in PBS and protected from light at rt.

### 2.7. Statistical Analysis and Validation

All fluorescence measurements were conducted in triplicate unless otherwise stated. The relative standard deviation (RSD) was calculated to assess the precision and reproducibility of the results. Statistical significance was determined where applicable using a threshold of *p* < 0.05. Calibration curve fitting and determination of detection limits are presented in the Section 3. To evaluate variability across concentration levels, a one-way ANOVA was performed, with statistical significance defined as *p* < 0.05. The corresponding RSD values and statistical outcomes are summarized in Table 1.

### 2.8. Calibrations

#### 2.8.1. Dual@nanodots

In a 48-well microplate, 100 µL of a suspension containing the dual@nanodots (1 mg mL^−1^ in PBS) was combined with CA19-9 standard solutions, spanning a concentration range from 4.98 × 10^−3^ to 8.39 × 10^2^ U mL^−1^. The total volume in each well was adjusted to 200 µL using PBS. All samples were prepared in triplicate (S/N = 3) and incubated for 20 min at rt. Following incubation, the suspensions were centrifuged at 4000 rpm for 5 min at 22 °C, and the supernatants were discarded. The resulting pellets were resuspended in 200 µL of PBS, and fluorescence measurements were taken relative to CA19-9 concentrations. Calibration curves were also established using 1% human serum diluted in PBS. Additional methodological details are provided in the Appendix A.

#### 2.8.2. Dual@nanoMIPs

100 µL of the dual@nanoMIPs suspension in PBS (1 mg mL^−1^) was pipetted into each well of a 48-well microplate and mixed with CA19-9 standard solutions ranging from 4.98 × 10^−3^ to 8.39 × 10^2^ U mL^−1^. The volume was adjusted to 200 µL per well using PBS. Triplicate samples (S/N = 3) were prepared and incubated for 20 min at rt. Following incubation, the mixtures were centrifuged at 4000 rpm for 5 min at 22 °C, and the supernatants were discarded. Pellets were resuspended in 200 µL of PBS, and fluorescence intensity was recorded across the range of CA19-9 concentrations. The same procedure was applied to the non-imprinted polymer controls (dual@nanoNIPs). Calibration experiments were also conducted in 1% human serum diluted in PBS. Additional experimental information is available in the Appendix A.

### 2.9. Selectivity Studies of the Dual@nanoMIPs

The selectivity of the dual@nanoMIPs for CA19-9 was evaluated using structurally and biologically relevant interferents. In a 48-well microplate, 100 µL of a 1 mg mL^−1^ dual@nanoMIPs suspension in PBS was incubated with 100 µL of CA19-9 at 100 U mL^−1^. For comparison, the same concentration of dual@nanoMIPs was incubated with individual solutions of the following non-target analytes in the absence of CA19-9: creatinine (CREA; 5 µg mL^−1^), matrix metalloproteinase-7 (MMP7; 5 ng mL^−1^), carcinoembryonic antigen (CEA; 5 ng mL^−1^), and human serum albumin (HSA; 10 µg mL^−1^). After incubation (20 min at rt, under shaking), the mixtures were centrifuged (4000 rpm, 5 min, 22 °C), and the pellets were resuspended in 200 µL of PBS for fluorescence measurement. The same protocol was applied to dual@nanoNIPs for baseline comparison. All experiments were performed in triplicate (S/N = 3).

### 2.10. Reproducibility and Stability of the Dual@nanoMIPs

Reproducibility was assessed using 1% human normal (HN) serum in PBS. Four concentrations of CA19-9 standards (0.749, 4.31, 24.2, and 141 U mL^−1^) were spiked into the serum solution, and 100 µL of dual@nanoMIPs suspension (1 mg mL^−1^) was added to each. The total volume in each well was adjusted to 200 µL. After incubation, the suspensions were centrifuged (4000 rpm, 5 min, 22 °C), supernatants were discarded, and pellets were resuspended in fresh 200 µL serum solution.

To evaluate the stability of the dual@nanoMIPs imprinted with CA19-9 (10 kU mL^−1^), fluorescence measurements were performed over time in PBS. A suspension of the dual@nanoMIPs (1 mg mL^−1^) was prepared, and 100 µL was dispensed into each well of a 48-well microplate. The final volume was brought to 200 µL with PBS, and fluorescence signals were recorded on days 0, 5, 10, and 20. Each measurement was carried out in triplicate (n = 3), and the relative standard deviation (RSD, %) was calculated to assess signal consistency. The same procedure was also applied to the corresponding non-imprinted controls, dual@nanoNIPs.

## 3. Results

### 3.1. Assembly Conditions of Dual@nanodots and of Dual@nanoMIPs

The assembly of dual@nanodots and their incorporation into dual@nanoMIPs was monitored via steady-state fluorescence spectroscopy. The y-QDs, surface-functionalized with 3-mercaptopropionic acid, exhibited an average emission maximum at 574.9 ± 1.9 nm with a quantum yield (QY) of 31.2 ± 2.3%, using rhodamine 6G as a reference standard (QY = 95%) [39,40]. The b-CDs, synthesized under mildly acidic conditions (pH 4.0), showed an emission peak at 466 nm and a QY of 22.6 ± 1.4%, using quinine sulfate as a standard (QY = 54%) [39,40]. Batch-to-batch reproducibility for both nanodots was consistent, with emission intensities varying by <7.5% and <6.2% for y-QDs and b-CDs, respectively (Appendix A). In the assembled dual@nanodots system, both emissions were present in PBS (10 mM, pH 7.4), with y-QDs dominating the fluorescence profile. When these nanodots were embedded within the polymer matrix during MIP synthesis, fluorescence quenching was observed, most notably in MIPs imprinted with 10 kU mL^−1^ of CA19-9 (Figure 2d). This effect was concentration-dependent, as lower imprinting (1 kU mL^−1^) resulted in less quenching (Figure 2b). The quenching behavior was partially reversible upon template removal using carbonate/bicarbonate buffer (10 mM, pH 9.8), indicating successful elution of CA19-9 from the polymer matrix. Only imprinted polymers exhibited this recovery (Figure 2d, green line)—no such change was observed in the non-imprinted controls (Figure 2c), underscoring the specificity of the binding. The significant fluorescence quenching in dual@nanoMIPs, particularly at higher CA19-9 imprinting concentrations, suggested the formation of specific binding sites that promoted close proximity between the QDs and the target antigen. While the mechanism of quenching was not conclusively resolved in this study, it is unlikely to be Förster resonance energy transfer (FRET), given the lack of spectral overlap between CA19-9 absorbance and QD emission (Appendix A). Instead, the data were consistent with a static or photoinduced electron transfer (PET) process, whereby excited-state electrons from QDs may be transferred to electron-deficient groups on CA19-9. However, this interpretation remains speculative in the absence of time-resolved photophysical data, and future studies using fluorescence lifetime spectroscopy are needed to validate this mechanism [42,43]. The partial fluorescence recovery observed after template removal was attributed to the loss of specific quenching interactions following CA19-9 elution. The absence of this recovery in the non-imprinted controls supports the formation of true imprinted sites rather than nonspecific adsorption. The effectiveness of surface imprinting was further evidenced by the absence of residual CA19-9 in the supernatants, as confirmed by UV-vis spectroscopy at 195 nm (Appendix A).

These results suggest that the imprinting concentration significantly influenced site density and sensor performance. The more pronounced quenching in the 10 kU mL^−1^ in the dual@nanoMIPs indicated improved rebinding due to increased site availability. Electrostatic interactions likely also contributed to this behavior, as CA19-9 is known to carry a net-negative charge under physiological conditions, promoting association with the positively charged functional monomers and the negatively charged surfaces of the y-QDs [44,45]. Lastly, the 1 h incubation time with CA19-9 was sufficient to reach equilibrium binding, as no further changes in fluorescence were observed with longer exposure (Figure 7a). This suggests that the dual@nanoMIPs exhibited rapid target recognition, a favorable property for diagnostic applications.

### 3.2. FTIR Analysis

In the FTIR spectra of the raw QDs, the presence of the OH stretching was evidenced by the broad peak at 3418 cm^−1^ (Figure 3, purple). The bands at 1564 cm^−1^ and 1402 cm^−1^ were attributed, respectively, to the symmetric and asymmetric stretching vibrations of the carboxylic groups of the surface of the yellow-emitting QDs, which were in the form of carboxylate anions COO^−^, attending the buffer used (PBS 10 mM, pH 7.4) [46]. In the spectrum of the carbon dots (Figure 3, green), the band at 3003 cm^−1^ relative to the CH stretching, the strong band at 1702 cm^−1^ correspondent to the C=O stretching, and the bands at 1558 cm^−1^ and at 1398 cm^−1^ were visible. These bands were also visible in the spectra of the dual@nanoMIPs (Figure 3, blue) and of the non-imprinted controls dual@nanoNIPs (Figure 3, brown), being associated, respectively, to the symmetric and asymmetric stretching of the carboxylic groups in the form of carboxylate anions COO^−^.

Relative to the dual@nanoMIPs, the band around 1103 cm^−1^ (Figure 3, blue) was attributed to the stretching vibrations of the C-O-C bonds from the polymeric matrix and was slightly visible in the spectra of the controls, dual@nanoNIPs (Figure 3, brown). In addition, the band at 986 cm^−1^ was attributed to the CH_2_ stretching of the polymer matrix, and the band at 1719 cm^−1^ was assigned to the C = O vibrations from the polymeric matrix (Figure 3, blue). The moderate band at 1154 cm^−1^ may be attributed to the C-H anti-symmetric stretching from CH_3_ groups from residues of antigen probably entrapped inside the polymer, as it also appeared in the spectrum of CA19-9 (Figure 3, pink) but was not visible in the spectrum of the dual@nanoNIPs (Figure 3 brown) [47]. Finally, the weak protuberance at 3280 cm^−1^ in the spectra of the CA19-9 (Figure 3, pink), which was slightly visible in the spectra of the dual@nanoMIPs (Figure 3 blue), was associated with N-H vibrations and may be related to antigen residues remaining upon washing, as it did not appear in the spectra of the dual@nanoNIPs.

### 3.3. SEM and EDS Analyses

The SEM and EDS analyses of the imprinted materials and their controls are shown in Figure 4. The dual@nanodots exhibited a spherical morphology; however, due to agglomeration in aqueous media, accurately determining the individual diameters of the nanoparticles, including the CDs and y-QDs, was challenging (Figure 4a). Both dual@nanoMIPs (Figure 4b) and dual@nanoNIPs (Figure 4c) exhibited agglomerated structures, with subtle differences between them. The dual@nanoMIPs displayed a more hollow structure compared to their non-imprinted counterparts. This distinction arose from the fabrication process: in dual@nanoMIPs, the removal of the antigen during polymerization created hollow binding sites within the matrix, corresponding to cavities previously occupied by antigen molecules. After template extraction, these sites remained vacant, resulting in a more porous structure, as observed in Figure 4b. In contrast, the absence of an antigen template in dual@nanoNIPs led to a more compact and dense structure, as evidenced by the SEM images in Figure 4c.

To enhance the clarity of the elemental composition data obtained by EDS, the atomic percentages of key elements identified in the spectra (Figure 4, z1–z3) were extracted and are summarized in Figure 4. As shown, carbon, oxygen, cadmium, and tellurium were consistently present across all three types of nanostructures. The dual@nanodots (z1) exhibited the highest levels of carbon, cadmium, and tellurium, likely due to greater surface exposure of these elements during EDS acquisition. In contrast, the dual@nanoMIPs (z2) displayed a lower carbon content than the dual@nanoNIPs (z3), which may reflect the removal of the template antigen and formation of a more porous structure. Additionally, the oxygen content was higher in the dual@nanoMIPs, possibly indicating the retention of trace antigen fragments within the polymer matrix after washing. These variations support successful imprinting and template removal processes and confirm the structural and compositional differences between imprinted and non-imprinted nanostructures.

### 3.4. Calibrations of Dual@nanoMIPs in PBS

The analytical data obtained during calibrations were analyzed according to the Stern–Volmer Equation (1), where I_0_ and I are the fluorescence intensities in the absence and in the presence of CA19-9, respectively, *k*_SV_ is the Stern–Volmer constant, and [Q] is the concentration of CA19-9 loaded on the imprinted rtIPs:I_0_/I = 1 + k_SV_ [Q](1)

A linear correlation in the fluorescence quenching (I_0_/I) of the dual@nanoMIPs was observed at the emission wavelength of the y-QDs (λ_em_ = 574.9 nm), but not at the emission wavelength of the b-CDs (λ_em_ = 467 nm), as seen in Figure 5b. This result was also observable to a lower extent with the calibrations of dual@nanoNIPs (Figure 5a), and even less with the dual@nanodots (Figure 5c). The corresponding Stern–Volmer plots were calculated using an average emission intensity value of the b-CDs during calibrations. These were applied as internal reference probes to account for variations caused by background matrix effects while remaining independent of the target concentration. Another parameter that was affected by the site accessibility and mass-transfer resistance of the biomolecule when working with imprinted polymers was the imprinting factor (IF), being evaluated by the following equation:IF = *k*_SV_ MIP/*k*_SV_ NIP (2)

It represents the capacity of the imprinted polymers to respond selectively to a given target, comparatively to the non-imprinted controls. Attending to the effect of the imprinting concentration, it was observed that a more pronounced decrease in the fluorescence signal during calibrations occurred with the use of the dual@nanoMIPs prepared from the imprinting concentration of 10 kU mL^−1^ of CA19-9 (Appendix A, blue) than with those prepared from 1 kU mL^−1^ imprinting of CA19-9 (Appendix A, pink). Taking into consideration these parameters, the dual@nanoMIPs prepared from 10 kU mL^−1^ imprinting of CA19-9 showed a LOD of 1.20 × 10^−3^ U mL^−1^ (S/N = 3), an IF of 3.04, and a *k*_SV_ of −0.0837 (Appendix A). The LOD value was calculated as the concentration needed to quench three times the standard deviation of the blank divided by the slope. The dual@nanodot probes were also calibrated, and a similar tendency was observed during calibrations with standards of CA19-9 prepared in the same interval range in PBS (10 mM, pH 7.4), but with slower kinetics. Effectively, the dual@nanodots showed a LOD of 4.36 × 10^−3^ U mL^−1^ (S/N = 3) and a *k*_SV_ of −0.0135, as shown in Appendix A.

### 3.5. Calibrations of Dual@nanoMIPs in Serum

To evaluate the sensor’s applicability in complex biological environments, additional calibration experiments were conducted using commercially available lyophilized human normal (HN) serum, commonly employed for quality control and diagnostic testing (Figure 6).

A 100-fold dilution was performed to adjust the matrix to conditions compatible with the sensor’s linear detection range and to minimize potential matrix effects, such as protein adsorption, viscosity, autofluorescence, and other nonspecific interactions that could compromise the sensitivity and specificity of fluorescence-based detection. HN serum was diluted to 1% in PBS to reduce fluorescence quenching and mitigate interference from endogenous serum components, thus enhancing the reliability of target detection. This dilution step is expected to remain necessary in future clinical applications to preserve sensor performance while maintaining biological relevance. The corresponding Stern–Volmer plots in this diluted matrix are presented in Appendix A. A linear correlation in fluorescence quenching (I_0_/I) was observed at the emission wavelength of the y-QDs (λ_em_ = 573.8 nm), while the reference b-CDs signal (λ_em_ = 466 nm) remained unaffected, confirming the ratiometric nature of the sensor. This linear behavior, comparable to that observed in PBS, suggests that the dual@nanoMIPs can function effectively even in diluted serum, although the use of undiluted or more complex fluids may still pose a challenge due to increased nonspecific interactions or optical interference. Notably, a more pronounced quenching effect was observed with dual@nanoMIPs imprinted at 10 kU mL^−1^ CA19-9 compared to those imprinted at 1 kU mL^−1^, demonstrating enhanced sensitivity (Appendix A). The calculated LOD in 1% serum for the 10 kU mL^−1^-imprinted dual@nanoMIPs was 2.40 × 10^−3^ U mL^−1^ (S/N = 3), with a k_SV_ of –0.0942 and an imprinting factor (IF) of 3.01. The sensor exhibited a linear response from 4.98 × 10^−3^ U mL^−1^ to 8.39 × 10^2^ U mL^−1^, encompassing the clinical diagnostic threshold for pancreatic cancer (CA19-9 > 37 U mL^−1^) [48]. By contrast, dual@nanodots showed lower sensitivity and a LOD of 3.97 × 10^−3^ U mL^−1^ with a smaller k_SV_ of –0.034, likely due to nonspecific adsorption of target molecules onto the negatively charged surface of y-QDs. While fluorescence quenching in controls indicated potential matrix-related interferences, the significantly higher response of imprinted materials demonstrated that the dual@nanoMIPs can effectively differentiate specific binding events from nonspecific interactions, even in a protein-rich environment like human serum. The matrix effect, although reduced through sample dilution, was further mitigated by the inclusion of an internal fluorescence reference (b-CDs), which compensated for fluctuations in signal intensity due to environmental variations, such as pH, ionic strength, or sample turbidity. This ratiometric design contributed to improved sensor accuracy and reliability in complex biological fluids. Moreover, the distinct fluorescence color gradient from yellow to green observed under UV light in 1% human serum, reinforced the dual@nanoMIPs’s potential for semi-quantitative visual detection. For future clinical applications, further optimization will be required for direct use in undiluted or minimally processed biological fluids, along with large-scale validation in patient-derived samples. Additionally, to improve visual consistency across samples with variable serum compositions, the use of ratiometric imaging filters or pre-set look-up tables (LUTs) is recommended. These tools can standardize fluorescence visualization by translating emission intensity ratios into uniform color representations, mitigating subtle spectral shifts caused by differences in serum background.

### 3.6. Reproducibility and Stability Studies of Dual@nanoMIPs

The dual@nanoMIPs demonstrated excellent analytical performance, with recovery values ranging from 99.86% to 107.25% and low RSDs between 0.84% and 1.90%, confirming high precision. As shown in Table 1, a statistically significant difference (*p* < 0.05) between the measured and spiked values was observed only at the lowest concentration (0.749 U/mL), likely reflecting greater variability near the detection limit. For all other concentrations, *p*-values > 0.05 confirmed that the measured results were statistically indistinguishable from the true values, supporting the accuracy and reliability of the sensor.

**Table 1 biomedicines-13-01629-t001:** Recovery, RSD, and statistical significance of dual@nanoMIPs in 1% HN serum spiked with CA19-9.

Sample (U/mL)	Spiked (U/mL)	Mean Found (U/mL)	RSD (%)	Recovery (%)	*p*-Value
1	0.749	0.803	1.90	107.25	0.0253
2	4.31	4.310	0.84	100.00	1.0000
3	24.2	24.167	1.67	99.86	0.8995
4	141.0	141.967	0.87	100.69	0.3078

In terms of stability, the dual@nanoMIPs retained 98.9 ± 2.1% of their initial fluorescence intensity after five days of storage in the dark, indicating excellent short-term photostability. The controls dual@nanoNIPs retained 97.3 ± 2.3% over the same period, within acceptable limits (Figure 7b). These findings underscore the potential of the dual@nanoMIPs platform as a reproducible and stable sensing tool for CA19-9 detection in biological samples.

### 3.7. Selectivity Studies of Dual@nanoMIPs

The selectivity tests shown in Figure 7c demonstrate that upon incubation of the dual@nanoMIPs with CA19-9 at 100 U mL^−1^ in PBS, the fluorescence intensity decreased to 73.7% of its initial value. In contrast, upon incubation of the interferent creatinine (CREA) at 5 µg mL^−1^, the fluorescence quenching was minimal, with only a 2.0% decrease. A similar trend was observed with matrix metalloproteinase 7 (MMP7) at 5 ng mL^−1^, which caused a deviation of 2.1% from the blank signal of the dual@nanoMIPs. Additionally, upon incubation with human serum albumin (HSA) at 10 µg mL^−1^, the fluorescence intensity increased by 1.6% relative to the blank signal. All deviations fell within the range of relative errors. The fluorescence enhancement caused by HSA may be attributed to its isoelectric point (~5.7), which allowed it to act as an electron donor on the surface of the yellow-emitting quantum dots (y-QDs), thereby increasing the fluorescence signal of the imprinted probes.

The CEA biomarker was used in this study, as having an average molecular weight of approximately 180 kDa, it could compete with the binding sites of the target CA19-9 (having approximately 210 kDa); nonetheless, it caused a negligible deviation on the fluorescence intensity of the dual@nanoMIPs to 96.6% of its initial value (Figure 7c). The same tendency was observed for other interfering biomolecules, such as CREA, MMP7, and HSA, suggesting that these molecules caused no interactions with the dual@nanoMIPs. As for the controls, they lacked specific recognition, as their fluorescence remained relatively unchanged regardless of the analyte present. These results confirm that the imprinting process successfully enhanced the binding specificity of the dual@nanoMIPs, making them a highly selective sensor for CA19-9 detection. Details on the protocol can be found in the SM file. While additional saccharides, such as glucose, mannose, and galactose, are recognized as potentially relevant interferents due to their structural similarity to glycan epitopes of CA19-9, their inclusion in this study was not feasible at this stage. However, these analytes are being considered for future selectivity assays to further validate the recognition specificity of the developed MIPs toward glycosylated biomolecules. The analytical response of the dual@nanoMIPs at 10 kU mL^−1^ for CA19-9 imprinting was compared with other probes reported in the literature for this target, as shown in Table 2.

Based on the results of Table 1, the dual@nanoMIPs developed in this work offered excellent performance, with a wide linear range and a low limit of detection comparable to or surpassing other detection methods for CA19-9. Their versatility, including usage in 1% human serum in PBS and its reliable colorimetric output, positions them as a valuable tool for clinical applications, particularly in monitoring pancreatic cancer.

## 4. Conclusions

This study reported the successful development of dual-emitting molecularly imprinted nanopolymers (dual@nanoMIPs) for the selective and sensitive detection of CA19-9, an important biomarker for pancreatic cancer. By integrating b-CDs as an internal reference and y-QDs as the responsive element, the sensor achieved reliable ratiometric fluorescence detection with a broad dynamic range (4.98 × 10^−3^ to 8.39 × 10^2^ U/mL) and strong selectivity in 1% human serum, displaying a visible color gradient form yellow to green under a 365 nm UV light. To enhance the practicability of visual readouts in real-world conditions, especially across patient samples with variable serum compositions, the use of ratiometric imaging filters or pre-set look-up tables (LUTs) is also recommended. These tools help standardize fluorescence visualization by converting emission intensity ratios into consistent color representations, thus minimizing the impact of spectral variations due to serum background. Nonetheless, further validation with clinical samples and comparison to gold-standard methods, such as ELISA, are necessary to confirm its clinical utility. Future work will focus on clinical evaluation, integration into diagnostic platforms, and stability assessments to advance toward commercialization.

## Figures and Tables

**Figure 1 biomedicines-13-01629-f001:**
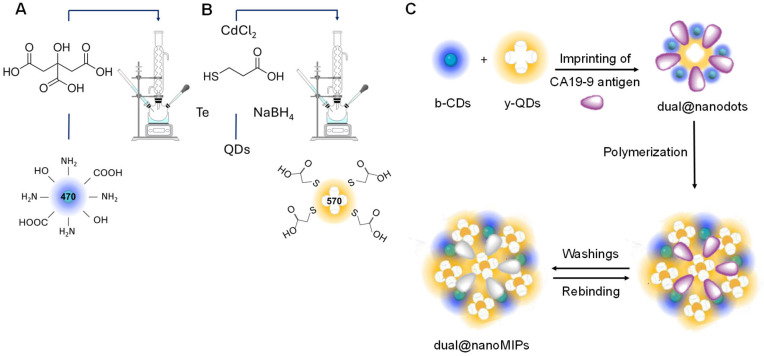
Synthesis of b-CDs (**A**) and of y-QDs (**B**). Assembly of the dual@nanoMIPs (**C**).

**Figure 2 biomedicines-13-01629-f002:**
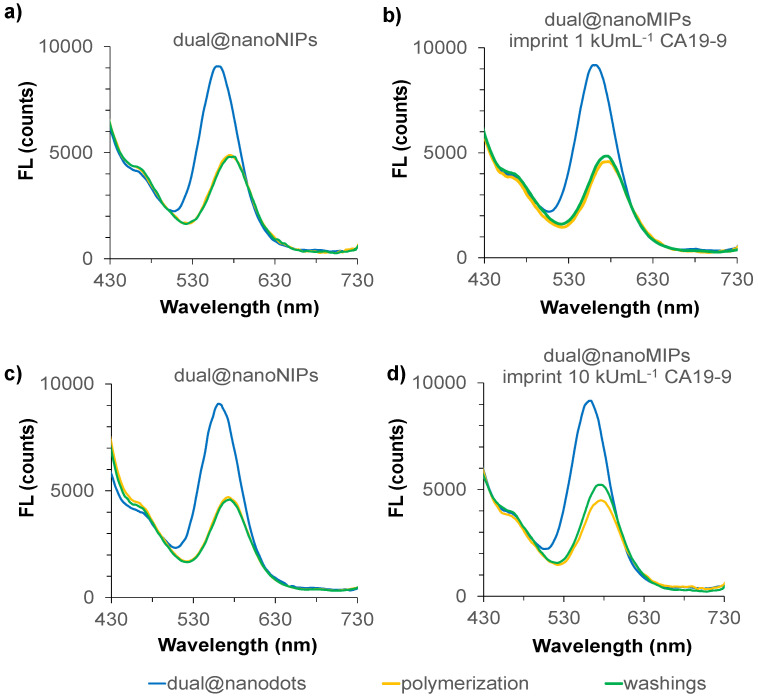
Fluorescence emissions obtained during assembly of the dual@nanoMIPs upon imprinting 1 kU/mL of CA19-9 (**b**) and 10 kU/mL of CA19-9 (**d**). The corresponding dual@nanoNIPs (**a**,**c**) in PBS.

**Figure 3 biomedicines-13-01629-f003:**
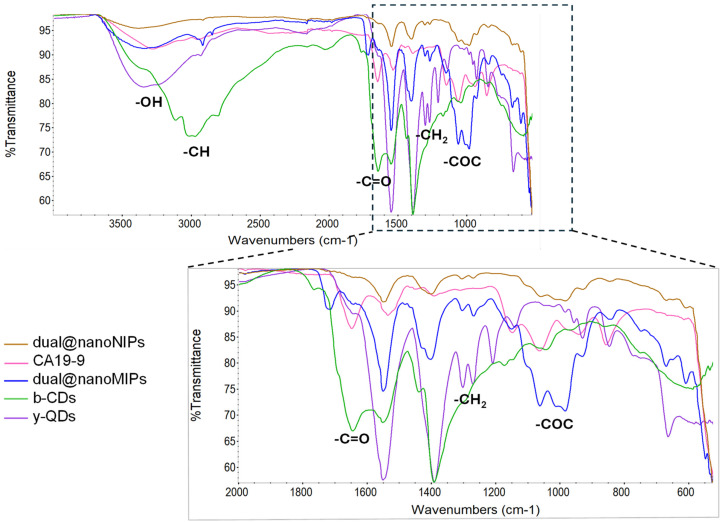
FTIR spectra of the dual@nanoNIPs (brown), CA19-9 (pink), dual@nanoMIPs (blue), b-CDs (green), and y-QDs (purple).

**Figure 4 biomedicines-13-01629-f004:**
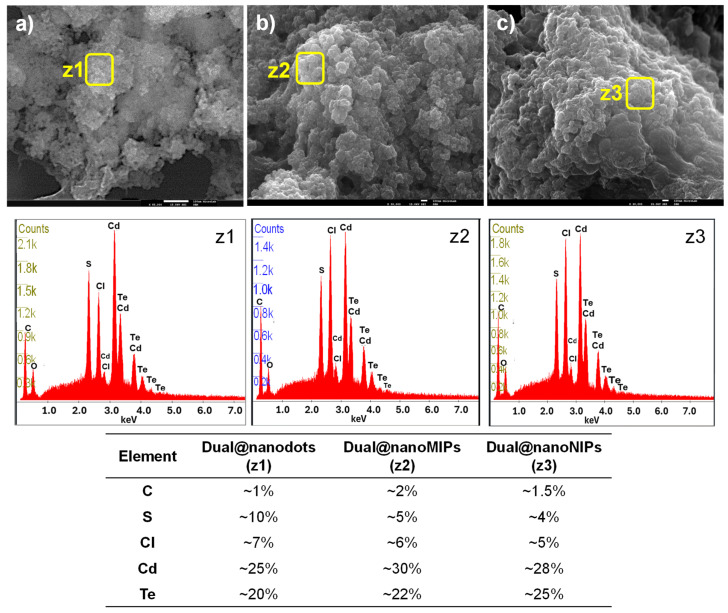
SEM images of (**a**) dual@nanodots, (**b**) dual@nanoMIPs after washings, and (**c**) dual@nanoNIPs after washings. The corresponding EDS analysis for dual@nanodots (**z1**), dual@nanoMIPs (**z2**), and dual@nanoNIPs (**z3**).

**Figure 5 biomedicines-13-01629-f005:**
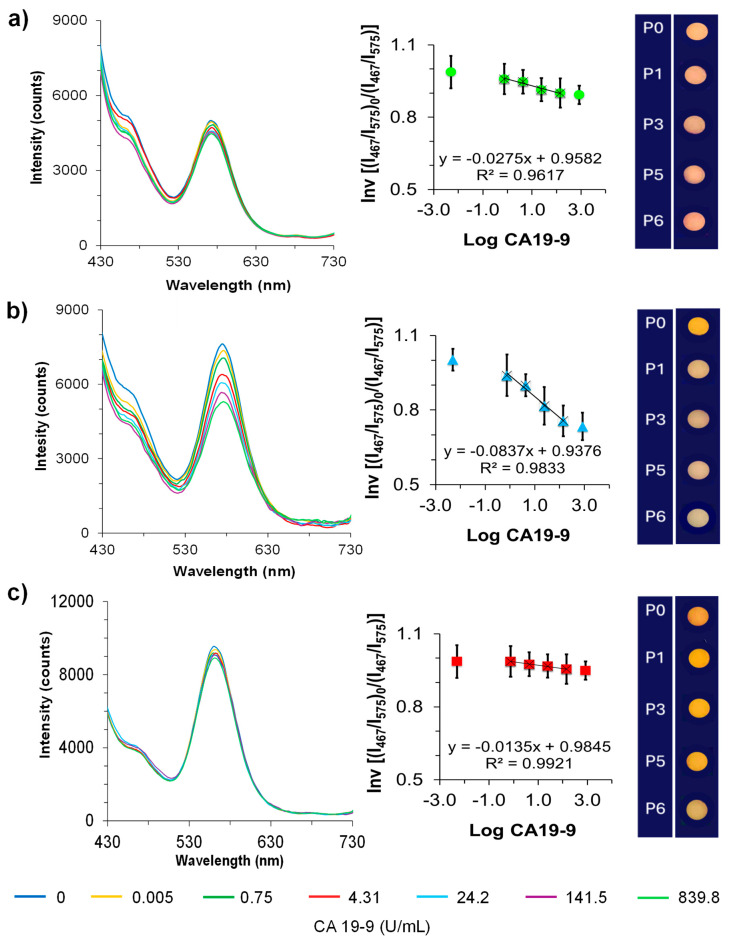
Fluorescence emissions and the corresponding gradient color panels observed under a 365 nm UV lamp and the corresponding emission spectra of the dual@nanoNIPs (**a**), dual@nanoMIPs (**b**), and dual@nanodots (**c**), during calibrations with CA19-9 standards in the interval range 4.98 × 10^−3^ U mL^−1^ to 8.39 × 10^2^ U mL^−1^ in PBS. Insets are the correspondent Stern–Volmer plots.

**Figure 6 biomedicines-13-01629-f006:**
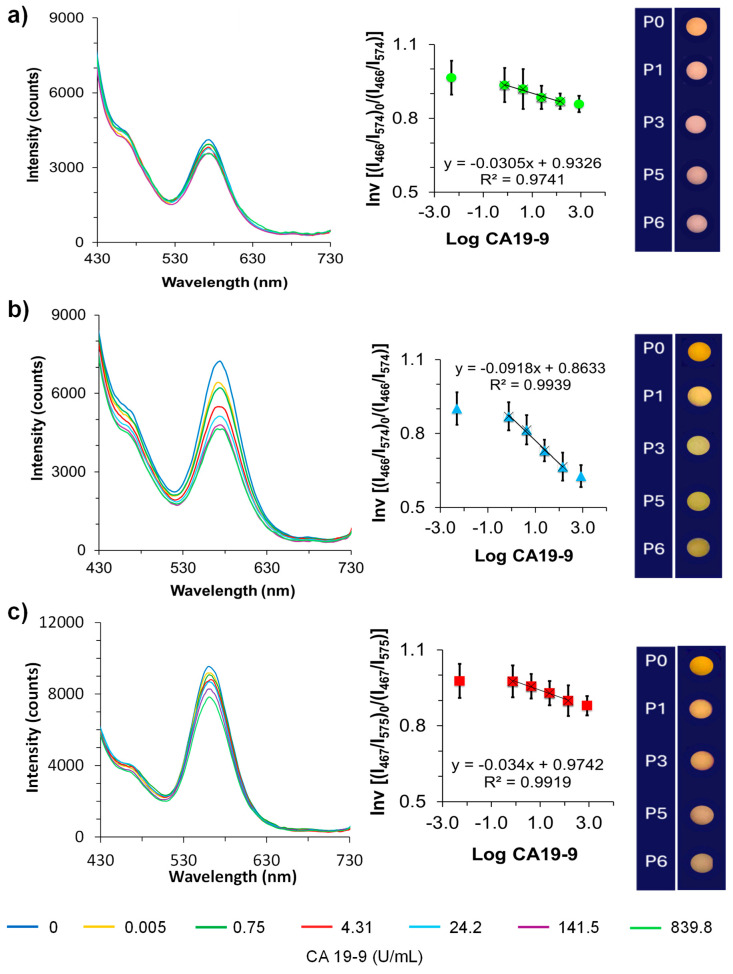
Fluorescence emissions and the corresponding gradient color panels observed under a 365 nm UV lamp and the corresponding emission spectra of the dual@nanoNIPs (**a**), dual@nanoMIPs (**b**), and dual@nanodots (**c**), during calibrations with CA19-9 standards in the interval range 4.98 × 10^−3^ U mL^−1^ to 8.39 × 10^2^ U mL^−1^ in 1% HN serum in PBS. Insets are the correspondent Stern–Volmer plots.

**Figure 7 biomedicines-13-01629-f007:**
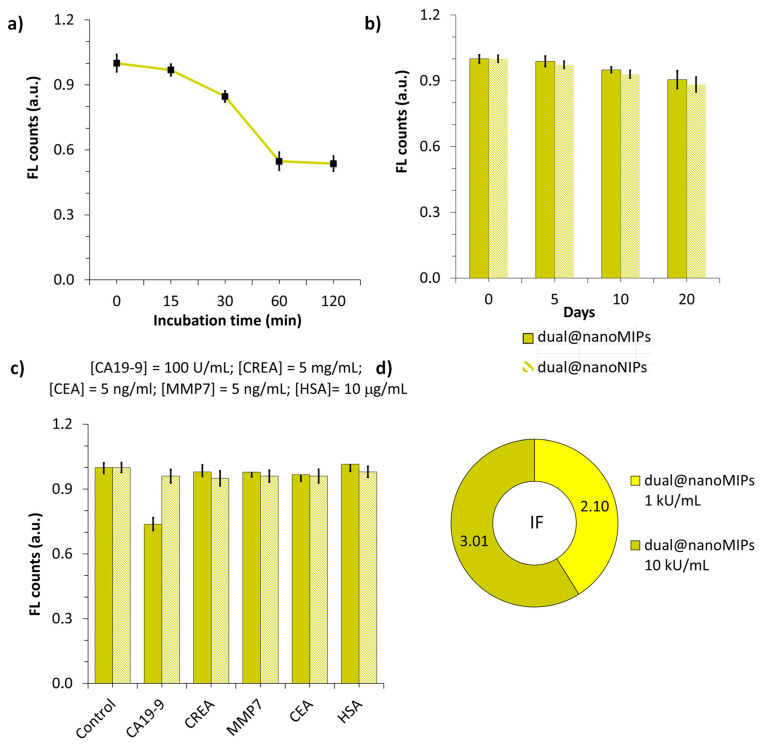
(**a**) Effect of the time of incubation of CA19-9 at 10 kU mL^−1^ in PBS during the assembly of dual@nanoMIPs, (**b**) stability of dual@nanoMIPs and its controls, after 20 days storage, rt, protected from light, (**c**) selectivity studies of dual@nanoMIPs and its controls in the presence of the interferents CREA, MMP7, CEA, and HSA, and (**d**) calculated imprinting factor of the dual@nanoMIPs prepared with two imprinting concentrations of CA19-9 (1 kU mL^−1^ and 10 kU mL^−1^) in 1% HN serum in PBS. Error bars represent standard deviation (S/N, n = 3).

**Table 2 biomedicines-13-01629-t002:** Comparison of the analytical response of the dual@nanoMIPs at 10 kU mL^−1^ imprinting of CA19-9, in 1% HN serum in PBS (10 mM, pH 7.4), with other works reported in the literature for the detection of CA19-9.

Method	Sensor type	Application	LR (U mL^−1^)	LOD (U mL^−1^)	Refs.
SERS	Immunosensor	PBS buffer	1 × 10^−5^−5 × 10^1^	3 × 10^0^	[49]
SPR	Graphene oxide immunosensor	Human saliva	0–2 × 10^−6^	1 × 10^−7^	[48]
CV/DPV	Electrochemical sensor	Diluted HS	1 × 10^−11^–1 × 10^−6^	6 × 10^−12^	[50,51]
SWV	Amperometric biosensor	HS	1 × 10^−13^–1 × 10^−7^	1.7 × 10^−14^	[52]
Au/Ag@SiO_2_FRET	Colorimetric	Cell and tissue human lines	1 × 10^−2^–1 × 10^−7^	2.2 × 10^−13^	[53]
Fluorescence	ULISA immunosensor		5 × 10^0^–2 × 10^3^	…	[14]
Fluorescence	dual@nanodots dual@nanoMIPs	1% HN serum in PBS	4.98 × 10^−3^–8.39 × 10^2^ 4.98 × 10^−3^–8.39 × 10^2^	3.97 × 10^−3^ 2.40 × 10^−3^	This Work

SPR: surface plasmon resonance; CV: cyclic voltammetry; DPV: differential pulse voltammetry; SWV: square wave voltammetry; FRET: fluorescence resonance energy transfer; ULISA: UCNP-linked immunosorbent assay; UCNP: up-conversion nanoparticles.

## Data Availability

All data supporting the reported results are included in the Appendix A of this article. No additional datasets were generated or analyzed beyond those presented in the manuscript and Appendix A.

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
