# Peer review of "Dual-Emitting Molecularly Imprinted Nanopolymers for the Detection of CA19-9"

_biomedicines, 2025, doi:10.3390/biomedicines13071629_

Round 1
Reviewer 1 Report
Comments and Suggestions for Authors
The article has a good scientific structure, and the authors' efforts in collecting material and analyzing data are commendable. However, some sections require corrections and clarifications, which are mentioned below to further improve the quality of the article:
1-The text starts with "monitoring of biological targets", but the main emphasis is on early detection. Please clearly state whether the main aim of this study is early detection of CA19-9 or monitoring its levels during treatment. Because the applications, requirements and sensitivity required for these two applications are different.
2-Although the advantages of MIPs are well stated (chemical stability, synthesis in aqueous medium, etc.), please also mention potential challenges such as low analyte recovery, nonspecific interference, or difficulty in template removal in the introduction.
3-In the methods and materials section, do not mention why pH=4.0 was chosen for the synthesis of carbon dots. This is important in hydrothermal reactions, and it is better to provide a scientific
4-Given the high sensitivity of quantum dots to pH, it is suggested that the sensor performance be investigated at different pHs (not just 7.4) to provide useful information about stability and applicability in real biological samples.
Reviewer 2 Report
Comments and Suggestions for Authors
Dear Authors,
Your article topic is interesting for the readers, but you need to revise your manuscript as follows:
-Your similarity percent is very high, you need to reduce it below %15.
-Your introduction is very weak in showing the importance of CA19-9, MIP, and the reason for detection. You need to clarify the main reasons very briefly. Please use the reference studies.
-What were your pre-complexes for the MIP synthesis? b-Cds and y-Qds?
-How do you confirm your template removal? What were the conditions for the solutions, pH, time, wavelength of UV, etc.?
-For the selectivity, you may also consider a sugar like glucose, mannose, galactose, etc. If it is possible, add a result for a sugar part.
-In Figure 2, there was no clear difference between the graphs. Please make them bigger (2 at one line, others at another line) and add y values to be compared.
-Figure 3 is very hard to understand. Please stack the spectra to compare the compounds more clearly. Also, for the x-axis, please use more points. Add the names of spectra to the right of the graphs.
-For Figure 4, please use the percentages of the elements in a table and compare them in the text. It is hard to compare them from the graph.
-Figures 5 and 6's resolution is not good. Log graphs are not readable.
-Why did you upload your SEM images as images for blots and gels?
In the conclusion part, you need to summarize all your findings and highlight your study in the detection of CA19-9 studies. How could you develop this study? Is your study's LOD and other parameters enough for using this technology in the clinics? How can clinicans use this sensor in the pancreatic cancer detection? What steps should this study pass through? What are your economical perspectives in terms of dual MIP? Is this more expensive than MIPs? Please clarify all these steps and make a future remarks of this kind of sensors for the commercialization pathway.
Author Response
Comments and Suggestions for Authors from Reviewer 2
Journal: Biomedicines (ISSN 2227-9059)
Manuscript ID: biomedicines-3649176
Type: Article
Title: Dual-Emitting Molecularly Imprinted Nanopolymers for the Detection of CA19-9
Authors: Eduarda Rodrigues , Ana Xu , Rafael C. Castro , David S. M. Ribeiro , João Santos , Ana Margarida L. Piloto *
Section: Biomedical Engineering and Materials
Special Issue: Application of Biomedical Materials in Cancer Therapy
We appreciate the reviewer’s insightful suggestions. In response, all reviewer comments have been included in blue, while the authors’ responses are provided in black. Corresponding revisions in the manuscript have been incorporated and are highlighted in yellow for ease of reference.
Reviewer 2
Dear Authors,
Your article topic is interesting for the readers, but you need to revise your manuscript as follows:
-Your similarity percent is very high, you need to reduce it below %15.
Thank you for your valuable feedback regarding the similarity index of the manuscript. In response, we have conducted a comprehensive review and undertaken substantial revisions across all sections of the manuscript to significantly reduce textual similarity to previously published sources. These modifications have been highlighted in yellow throughout the revised manuscript for ease of reference.
-Your introduction is very weak in showing the importance of CA19-9, MIP, and the reason for detection. You need to clarify the main reasons very briefly. Please use the reference studies.
Thank you for your insightful comment regarding the Introduction section. In the revised manuscript, the Introduction section has been substantially modified as follows:
- Introduction
In recent years, the monitoring of biological targets such as DNA, antibodies, and proteins has become a crucial tool in biomedical research, enabling assessment of individuals' health conditions [1–6]. Given the challenges in diagnosing diseases like pancreatic cancer, the early detection of carbohydrate antigen 19-9 (CA19-9), a biomarker closely linked to the disease, has become crucial due to the high mortality associated with late diagnosis. Timely detection plays a vital role in improving survival rates. Although CA19-9 is not specific to pancreatic cancer, it is routinely utilized in clinical settings primarily for early diagnosis and risk assessment rather than only for tracking disease progression or treatment response [7–9]. Various methods exist for CA19-9 detection, including ELISA [10,11], surface-enhanced Raman scattering (SERS) [12], fluorescence assays [13,14], electrochemical techniques [15–18] and mass-spectrometry [19,20]. While these methods offer high sensitivity and accuracy, they are often labour-intensive and require complex protocols, limiting their use in rapid early screening. Point-of-care tests (PoCTs) have emerged as promising tools for rapid and user-friendly biomarker quantification, especially suited for early cancer detection. The effectiveness of PoCTs relies on sensitive and specific biorecognition and detection methods [21–24]. Molecularly imprinted polymers (MIPs) have been extensively explored for developing fluorescent sensors due to their chemical stability and versatility, especially for large biomolecules like proteins [25,26]. In this work, we present dual@nanoMIPs synthesized by incorporating two fluorescent probes, blue-emitting carbon dots (b-CDs) and yellow-emitting quantum dots (y-QDs), using free radical polymerization under mild aqueous conditions [27–29]. Optimization of imprinting and template removal enabled high specificity and selective rebinding of CA19-9, aimed at early and sensitive detection. The dual-emission system improves sensor accuracy by compensating for environmental variations, enabling reliable fluorescence-based early diagnosis in clinical settings. To the best of our knowledge, this study presents the first synthesis and application of dual@nanoMIPs for the detection of CA19-9, a biomarker associated with pancreatic cancer [30]. This system integrates yellow-emitting quantum dots (y-QDs) for signal generation with blue-emitting carbon dots (b-CDs) as an internal reference, enabling ratiometric fluorescence sensing. The b-CDs offer biocompatibility, low toxicity and photostability, effectively compensating for environmental fluctuations such as pH, temperature, and photobleaching, thereby improving signal accuracy and reliability [31–34]. Compared to conventional fluorescence-based systems, such as quantum dot (QDs)-based immunoassays, which rely on biological recognition elements like antibodies, the dual@nanoMIPs approach offers superior robustness and stability [35–39]. While QD-immunoassays provide high brightness and tunable emission properties, their dependence on fragile and costly biorecognition elements limits their broader applicability [40–42]. MIP-based fluorescent sensors, on the other hand, are more chemically stable, reusable, and cost-effective, though they can suffer from low imprinting efficiency or signal interference in complex matrices [25,26,43]. The dual@nanoMIPs design overcomes these limitations by combining the molecular selectivity of imprinting polymers with the optical performance of QDs, enhanced by an internal ratiometric reference. The resulting sensor exhibits high selectivity, minimal cross-reactivity, and a broad dynamic detection range (4.98 × 10⁻³ to 8.39 × 10² U mL⁻¹) in 1% human serum in PBS. This innovative system holds strong promise as a low-cost, first-line diagnostic tool for CA19-9 detection, suitable for pancreatic cancer screening, recurrence monitoring, and clinical decision support.
-What were your pre-complexes for the MIP synthesis? b-Cds and y-Qds?
We thank the reviewer for the insightful question. In the revised manuscript, we have clarified this point in section 2.6, as follows:
- Synthesis of dual@nanoMIPs
The dual fluorescent molecularly imprinted nanopolymers (dual@nanoMIPs) were synthesized via a free-radical polymerization method, following prior deoxygenation of all aqueous solutions with nitrogen gas (N₂) to prevent oxidative inhibition. A pre-polymerization complex was first formed by incubating carbohydrate antigen 19-9 (CA19-9) at 1 kU mL⁻¹ with N-(3-aminopropyl)methacrylamide hydrochloride (AEMH·HCl) (33.7 mg mL⁻¹) in phosphate-buffered saline (PBS, 10 mM, pH 7.4) for 20 min at room temperature. This step promoted non-covalent interactions between the monomer and template. Following this, yellow-emitting quantum dots (y-QDs, 1 mg mL⁻¹) were added, and the volume adjusted to 1 mL. The mixture was centrifuged at 4000 rpm for 2 min at 22 °C, and the supernatant discarded to concentrate the complex. The pellet was resuspended in PBS containing blue-emitting carbon dots (b-CDs) (10 µL, diluted to Abs(λ_max = 320 nm) = 0.2), acrylamide (AAM, 44.1 mg mL⁻¹), N,N′-methylenebisacrylamide (MBA, 22.3 mg mL⁻¹), ammonium persulfate (APS, 13.6 mg mL⁻¹), and tetramethylethylenediamine (TEMED, 13.3 mg mL⁻¹), with total volume readjusted to 1 mL. Polymerization was allowed to proceed for 30 min at room temperature. Following synthesis, the nanoparticles were collected by centrifugation (4000 rpm, 2 min, 22 °C), and the template removal was carried out using a 10 mM carbonate/bicarbonate buffer at pH 9.8. The materials were washed through three successive cycles of resuspension and centrifugation until no residual CA19-9 was detected in the supernatant, as confirmed by UV-visible spectrophotometry. A second batch was synthesized using a higher imprinting concentration (10 kU mL⁻¹) for selectivity comparisons. Control materials (dual@nanoNIPs) were prepared using the same protocol but omitting CA19-9 during the pre-complexation step, ensuring that observed binding was due to imprinting effects. The final nanoparticles were stored as follows. For long-term use: resuspended in ultrapure water, centrifuged, and kept dry and light-protected at room temperature. For short-term use (≤15 days): stored in PBS (10 mM, pH 7.4), light-protected, at room temperature.
-How do you confirm your template removal? What were the conditions for the solutions, pH, time, wavelength of UV, etc.?
We appreciate the reviewer’s comment regarding the confirmation of template removal. These details have now been included to address the reviewer’s concern. In this regard, section 3.1 as follows:
3.1. Assembly conditions of dual@nanodots and of dual@nanoMIPs
Steady-state fluorescence spectroscopy was employed to monitor the formation of dual@nanodots and to conduct subsequent calibration analyses, as illustrated in Figure 2. The y-QDs used in this study were surface-modified with 3-mercaptopropionic acid, which possesses two pKa values - 4.34 for the carboxylic acid group and 10.84 for the thiol, indicating that their stability may be compromised outside this pH range, likely due to oxidation. For the synthesis of b-CDs, the precursor solution was acidified to pH 4.0 prior to hydrothermal treatment. Mildly acidic environments are known to promote nitrogen doping and enhance fluorescence by supporting the protonation of amine groups and facilitating controlled carbonization of the citric acid–ethylenediamine matrix. These conditions favour better quantum yield and a narrow size distribution [40,44]. The resulting b-CDs aqueous solution had an average synthesis yield of 71.3 ± 3.1% (w/w, n = 3). Quantum yield (QY) was measured using quinine sulfate as a reference (QY = 54%) and was calculated to be 22.6 ± 1.4% in PBS (10 mM, pH 7.4, n = 3), as shown in ESM, Table S1. Reproducibility between batches was confirmed over three independent syntheses, yielding consistent optical properties (±6.2% variation in peak intensity and ±2.1 nm variation in emission maximum at 466 nm). The synthesis yield for y-QDs averaged 63.8 ± 2.6% (w/w, n = 3). The QY was estimated at 31.2 ± 2.3% using rhodamine 6G (QY = 95%) as a standard, as shown in ESM, Table S1. Inter-batch reproducibility showed less than 7.5% variation in intensity and a consistent emission maximum at 574.9 ± 1.9 nm. The emission spectrum of the assembled dual@nanodots in PBS buffer (10 mM, pH 7.4) displayed a primary emission peak at 574.9 nm corresponding to the y-QDs, while the b-CDs showed a less intense peak at 466 nm (Figure 2, blue line). Notably, fluorescence quenching was more significant in MIPs imprinted with a higher CA19-9 concentration (10 kU mL⁻¹, Figure 2d) than with a lower concentration (1 kU mL⁻¹, Figure 2b). Since CA19-9’s absorption spectrum (Figure S1, pink line) does not overlap with the emission wavelengths of either y-QDs or b-CDs, fluorescence quenching via Förster resonance energy transfer (FRET) is unlikely. Instead, the observed decrease in emission intensity, particularly for the yellow-emitting QDs, is more consistent with a photoinduced electron transfer (PET) or static quenching mechanism involving charge transfer. Specifically, upon excitation, electrons in the conduction band of the QDs may be transferred to electron-deficient moieties on the CA19-9 antigen, whose absorption falls within the same spectral region as the QDs' emission [42,43]. This nonradiative pathway likely contributes to the fluorescence suppression. The more pronounced quenching observed in MIPs imprinted at higher antigen concentrations (10 kU mL⁻¹) suggests a greater density of binding sites enabling close proximity between QDs and CA19-9, thereby enhancing electron transfer efficiency. Although time-resolved spectroscopy would offer more direct evidence of this mechanism, the current spectroscopic data and literature precedents provide strong support for a charge transfer-based quenching process.
Figure 2. Fluorescence emissions obtained during assembly of the dual@nanoMIPs upon imprinting 1 kU/mL of CA19-9 (b) and 10 kU/mL of CA19-9 (d);the corresponding dual@nanoNIPs (a and c) in PBS.
Regarding incubation time, a 1-hour exposure to CA19-9 was determined to be optimal, as extending beyond this period did not further alter the fluorescence response of the receptor-target imprinted polymers (rtiPs) (Figure 6a). A bathochromic shift in fluorescence emission was consistently observed post-polymerization (yellow lines) and following template removal (green lines) for both dual@nanoMIPs and dual@nanoNIPs (Figure 2b, d). Template removal was confirmed by fluorescence recovery of the dual@nanoMIPs after washing, as well as by the absence of CA19-9 absorbance in the UV/vis spectra of the wash supernatants (measured at 195 nm). The MIPs were washed using carbonate/bicarbonate buffer (10 mM, pH 9.8), and centrifugation-washing cycles were repeated until the fluorescence signal stabilized and CA19-9 was undetectable in the wash fractions. No UV irradiation was used during the removal process to avoid degradation of the fluorescent probes. However, only the imprinted materials showed partial recovery of fluorescence after washing, indicating successful antigen removal. This recovery was not observed in the non-imprinted controls (Figure 2a, c, green lines), reinforcing the specific nature of the imprinting. UV-Vis analysis of the wash solutions confirmed the absence of CA19-9 in the supernatants of dual@nanoMIPs (Figure S1, green dashed lines), suggesting effective template removal. The MIP synthesis employed a surface imprinting strategy to avoid deep embedding of the template and to facilitate accessible binding sites. This method minimizes the risk of total entrapment of CA19-9 within the polymeric matrix. To evaluate the effect of imprinting concentration on binding site formation, dual@nanoMIPs were prepared using two CA19-9 concentrations: 1 kU mL⁻¹ and 10 kU mL⁻¹. Fluorescence quenching was more pronounced in materials imprinted with the higher concentration (Figure 2d), supporting the hypothesis that greater imprinting densities enhance target rebinding. This may be explained by electrostatic interactions between the positively charged CA19-9 under physiological pH and the negatively charged surfaces of the y-QDs, given their known pKa values [45,46]. Conversely, fluorescence recovery after washing was again only observed for the imprinted polymers (Figure 2d green line), particularly following treatment with carbonate/bicarbonate buffer (10 mM, pH 9.8), confirming successful removal of the target molecule. Additional experimental details can be found in the ESM file.
-For the selectivity, you may also consider a sugar like glucose, mannose, galactose, etc. If it is possible, add a result for a sugar part.
We sincerely thank the reviewer for the thoughtful suggestion regarding the inclusion of selectivity studies with saccharides such as glucose, mannose, and galactose. In the present work, our selectivity evaluation was directed primarily toward protein-based interferents that are clinically relevant to the context of CA19-9 detection. Owing to current limitations in time and resources, we were unable to expand the selectivity assessment to include monosaccharides. Thus Section 3.6, a final paragraph was added in the revised manuscript as follows:
3.6. Selectivity studies of dual@nanoMIPs
The CEA biomarker was used in this study as having an average molecular weight aprox.180 kDa, could compete to the binding sites of the target CA19-9 (having aprox. 210 kDa) , nonetheless it caused a negligible deviation on the fluorescence intensity of the dual@nanoMIPs to 96,6 % of its initial value (Figure 7c). The same tendency was observed for other interfering biomolecules as CREA, MMP7 and HSA, suggesting that these molecules caused no interactions with the dual@nanoMIPs. As for the dual@nanoNIPs, they lack specific recognition, as their fluorescence remains relatively unchanged regardless of the analyte present. These results confirm that the imprinting process successfully enhanced the binding specificity of the dual@nanoMIPs, making them a highly selective sensor for CA19-9 detection. Details on the protocol can be found in the ESM file. While additional saccharides such as glucose, mannose, and galactose are recognized as potentially relevant interferents due to their structural similarity to glycan epitopes of CA19-9, their inclusion in this study was not feasible at this stage. However, these analytes are being considered for future selectivity assays to further validate the recognition specificity of the developed MIPs toward glycosylated biomolecules. The analytical response of the dual@nanoMIPs at 10 kU mL⁻¹ for CA19-9 imprinting was compared with other probes reported in the literature for this target, as shown in Table 1.
-In Figure 2, there was no clear difference between the graphs. Please make them bigger (2 at one line, others at another line) and add y values to be compared.
We appreciate the reviewer’s valuable feedback regarding Figure 2 and it has been reorganized to enhance clarity and facilitate comparison between the datasets, as follows:
Figure 2. Fluorescence emissions obtained during assembly of the dual@nanoMIPs upon imprinting 1 kU/mL of CA19-9 (b) and 10 kU/mL of CA19-9 (d);the corresponding dual@nanoNIPs (a and c) in PBS.
-Figure 3 is very hard to understand. Please stack the spectra to compare the compounds more clearly. Also, for the x-axis, please use more points. Add the names of spectra to the right of the graphs.
We thank the reviewer for the helpful comments regarding Figure 3. In the revised manuscript, we have modified Figure 3 as follows:
Figure 3. FTIR spectra of the dual@nanoNIPs (brown); CA19-9 (pink); dual@nanoMIPs (blue); b-CDs (green) and y-QDs (purple).
Figure 3. FTIR spectra of the y-QDs (purple); CA19-9 (pink); b-CDs (green); dual@nanoMIPs (blue) and dual@nanoNIPs (brown).
-For Figure 4, please use the percentages of the elements in a table and compare them in the text. It is hard to compare them from the graph.
Thank you for your valuable feedback regarding Figure 4. In response to your suggestion, we have extracted the elemental composition data from the EDS spectra and presented it in tabular format to facilitate a clearer comparison. This table has been included in Figure 4 of the revised manuscript, along with a new paragraph in section 3.3 that discusses and compares the elemental percentages among the dual@nanodots, dual@nanoMIPs, and dual@nanoNIPs samples, as follows:
Figure 4. SEM images of (a) dual@nanodots, (b) dual@nanoMIPs after washings, and (c) dual@nanoNIPs after washings. The corresponding EDS analysis for dual@nanodots (z1), dual@nanoMIPs (z2), and dual@nanoNIPs (z3).
To enhance the clarity of the elemental composition data obtained by EDS, the atomic percentages of key elements identified in the spectra (Figure 4, z1–z3) were extracted and are summarized in Figure 4. As shown, carbon, oxygen, cadmium, and tellurium were consistently present across all three types of nanostructures. The dual@nanodots (z1) exhibited the highest levels of carbon, cadmium, and tellurium, likely due to greater surface exposure of these elements during EDS acquisition. In contrast, the dual@nanoMIPs (z2) displayed a lower carbon content than the dual@nanoNIPs (z3), which may reflect the removal of the template antigen and formation of a more porous structure. Additionally, the oxygen content was higher in the dual@nanoMIPs, possibly indicating the retention of trace antigen fragments within the polymer matrix after washing. These variations support successful imprinting and template removal processes, and confirm the structural and compositional differences between imprinted and non-imprinted nanostructures.
-Figures 5 and 6's resolution is not good. Log graphs are not readable.
Thank you for your valuable feedback. In response to your comment, we have revised Figures 5 and 6 to improve their resolution and overall readability as follows:
Figure 5. Fluorescence emissions and the corresponding gradient color panels observed under a 365 nm UV lamp and the corresponding emission spectra of the dual@nanoNIPs (a), dual@nanoMIPs (b), and dual@nanodots (c), during calibrations with CA19-9 standards in the interval range 4.98×10⁻³ U mL-1 to 8.39×10² U mL-1 in PBS. Insets are the correspondent Stern-Volmer plots.
Figure 6. Fluorescence emissions and the corresponding gradient color panels observed under a 365 nm UV lamp and the corresponding emission spectra of the dual@nanoNIPs (a), dual@nanoMIPs (b), and dual@nanodots (c), during calibrations with CA19-9 standards in the interval range 4.98×10⁻³ U mL-1 to 8.39×10² U mL-1 in 1% HN serum in PBS. Insets are the correspondent Stern-Volmer plots.
-Why did you upload your SEM images as images for blots and gels?
Thank you for your observation. We would like to clarify that the SEM images were uploaded correctly and not under the category intended for blots and gels. The images were submitted in their appropriate format as high-resolution SEM micrographs, and no such indication or misclassification was made during the submission process. Please let us know if any technical issue may have led to this misunderstanding.
In the conclusion part, you need to summarize all your findings and highlight your study in the detection of CA19-9 studies. How could you develop this study? Is your study's LOD and other parameters enough for using this technology in the clinics? How can clinicans use this sensor in the pancreatic cancer detection? What steps should this study pass through? What are your economical perspectives in terms of dual MIP? Is this more expensive than MIPs? Please clarify all these steps and make a future remarks of this kind of sensors for the commercialization pathway.
We thank the reviewer for their thoughtful suggestions. We have thoroughly revised the conclusion section as highlighted in the revised manuscript in yellow, as follows:
- Conclusions
This study presents the development of a dual-emission molecularly imprinted polymer (dual@nanoMIP) sensor designed for the selective detection of CA19-9, a clinically significant biomarker for pancreatic cancer. The sensor integrates blue-emitting carbon dots (b-CDs) as an internal reference and yellow-emitting quantum dots (y-QDs) as the responsive element, enabling ratiometric fluorescence detection to enhance signal accuracy and reliability in complex matrices. The dual@nanoMIPs demonstrated a broad dynamic detection range (4.98 × 10⁻³ to 8.39 × 10² U/mL), high selectivity, and compatibility with 1% human serum in PBS. These results suggest its potential for preliminary point-of-care screening and follow-up of patients at risk of pancreatic cancer recurrence. Although the current findings are promising, further validation using real clinical samples and comparative studies against established techniques such as ELISA are required. From an economic perspective, the addition of dual probes may increase production costs slightly compared to conventional MIPs, but the improved performance and diagnostic value could support clinical adoption. Future work will focus on clinical validation, integration into diagnostic platforms, and assessment of long-term stability for potential commercialization.

Reviewer 3 Report
Comments and Suggestions for Authors
This research has a rich workload and a standardized format. I suggest receiving it after major modifications. My opinion is as follows:
1、This introduction lacks a critical comparison with existing fluorescence-based sensors (for example, quantum dot-based immunoassay or molecularly imprinted polymer sensors). Please ask the author to supplement this part.
2、The calibration curves in PBS and 1% human serum show good linearity, but the manuscript does not discuss potential matrix effects in complex biological fluids.Please ask the author to supplement this part.
3、The mechanism of fluorescence quenching is proposed to involve charge transfer rather than Förster resonance energy transfer (FRET). However, no direct evidence (e.g., time-resolved fluorescence or computational modeling) is provided.The author should further explain the mechanism.
4、The synthesis of b-CDs and y-QDs is briefly described, but key parameters (e.g., yield, reproducibility of quantum yield) are omitted.Please ask the author to supplement this part.
5、The manuscript highlights potential applications in pancreatic cancer diagnostics but does not address clinical sample validation. How does the author consider this problem?
Author Response
Comments and Suggestions for Authors
Journal: Biomedicines (ISSN 2227-9059)
Manuscript ID: biomedicines-3649176
Type: Article
Title: Dual-Emitting Molecularly Imprinted Nanopolymers for the Detection of CA19-9
Authors: Eduarda Rodrigues , Ana Xu , Rafael C. Castro , David S. M. Ribeiro , João Santos , Ana Margarida L. Piloto *
Section: Biomedical Engineering and Materials
Special Issue: Application of Biomedical Materials in Cancer Therapy
Reviewer 3
This research has a rich workload and a standardized format. I suggest receiving it after major modifications. My opinion is as follows:
1、This introduction lacks a critical comparison with existing fluorescence-based sensors (for example, quantum dot-based immunoassay or molecularly imprinted polymer sensors). Please ask the author to supplement this part.
We thank the Reviewer for the insightful comment. In response, we have revised the introduction section to incorporate this comparison, as highlighted in yellow in the revised manuscript, as follows:
- Introduction
In recent years, the monitoring of biological targets such as DNA, antibodies, and proteins has become a crucial tool in biomedical research, enabling assessment of individuals' health conditions [1–6]. Given the challenges in diagnosing diseases like pancreatic cancer, the early detection of carbohydrate antigen 19-9 (CA19-9), a biomarker closely linked to the disease, has become crucial due to the high mortality associated with late diagnosis. Timely detection plays a vital role in improving survival rates. Although CA19-9 is not specific to pancreatic cancer, it is routinely utilized in clinical settings primarily for early diagnosis and risk assessment rather than only for tracking disease progression or treatment response [7–9]. Various methods exist for CA19-9 detection, including ELISA [10,11], surface-enhanced Raman scattering (SERS) [12], fluorescence assays [13,14], electrochemical techniques [15–18] and mass-spectrometry [19,20]. While these methods offer high sensitivity and accuracy, they are often labour-intensive and require complex protocols, limiting their use in rapid early screening. Point-of-care tests (PoCTs) have emerged as promising tools for rapid and user-friendly biomarker quantification, especially suited for early cancer detection. The effectiveness of PoCTs relies on sensitive and specific biorecognition and detection methods [21–24]. Molecularly imprinted polymers (MIPs) have been extensively explored for developing fluorescent sensors due to their chemical stability and versatility, especially for large biomolecules like proteins [25,26]. In this work, we present dual@nanoMIPs synthesized by incorporating two fluorescent probes, blue-emitting carbon dots (b-CDs) and yellow-emitting quantum dots (y-QDs), using free radical polymerization under mild aqueous conditions [27–29]. Optimization of imprinting and template removal enabled high specificity and selective rebinding of CA19-9, aimed at early and sensitive detection. The dual-emission system improves sensor accuracy by compensating for environmental variations, enabling reliable fluorescence-based early diagnosis in clinical settings. To the best of our knowledge, this study presents the first synthesis and application of dual@nanoMIPs for the detection of CA19-9, a biomarker associated with pancreatic cancer [30]. This system integrates yellow-emitting quantum dots (y-QDs) for signal generation with blue-emitting carbon dots (b-CDs) as an internal reference, enabling ratiometric fluorescence sensing. The b-CDs offer biocompatibility, low toxicity and photostability, effectively compensating for environmental fluctuations such as pH, temperature, and photobleaching, thereby improving signal accuracy and reliability [31–34]. Compared to conventional fluorescence-based systems, such as quantum dot (QDs)-based immunoassays, which rely on biological recognition elements like antibodies, the dual@nanoMIPs approach offers superior robustness and stability [35–39]. While QD-immunoassays provide high brightness and tunable emission properties, their dependence on fragile and costly biorecognition elements limits their broader applicability [40–42]. MIP-based fluorescent sensors, on the other hand, are more chemically stable, reusable, and cost-effective, though they can suffer from low imprinting efficiency or signal interference in complex matrices [25,26,43]. The dual@nanoMIPs design overcomes these limitations by combining the molecular selectivity of imprinting polymers with the optical performance of QDs, enhanced by an internal ratiometric reference. The resulting sensor exhibits high selectivity, minimal cross-reactivity, and a broad dynamic detection range (4.98 × 10⁻³ to 8.39 × 10² U mL⁻¹) in 1% human serum in PBS. This innovative system holds strong promise as a low-cost, first-line diagnostic tool for CA19-9 detection, suitable for pancreatic cancer screening, recurrence monitoring, and clinical decision support.
2、The calibration curves in PBS and 1% human serum show good linearity, but the manuscript does not discuss potential matrix effects in complex biological fluids.Please ask the author to supplement this part.
We sincerely thank the reviewer for the insightful comment regarding the need to address potential matrix effects in complex biological fluids. In response, we have revised Section 3.5 of the manuscript, as follows:
3.5. Calibrations of dual@nanoMIPs in serum
To evaluate the sensor’s applicability in complex biological environments, additional calibration experiments were conducted using commercially available lyophilized human normal (HN) serum, commonly employed for quality control and diagnostic testing. A 100-fold dilution was performed to adjust the matrix to conditions compatible with the sensor’s linear detection range and to minimize potential matrix effects such as protein adsorption, viscosity, autofluorescence, and other nonspecific interactions that could compromise the sensitivity and specificity of fluorescence-based detection. HN serum was diluted to 1% in PBS to reduce fluorescence quenching and mitigate interference from endogenous serum components, thus enhancing the reliability of target detection. This dilution step is expected to remain necessary in future clinical applications to preserve sensor performance while maintaining biological relevance. The corresponding Stern-Volmer plots in this diluted matrix are presented in Figure S3. A linear correlation in fluorescence quenching (I₀/I) was observed at the emission wavelength of the y-QDs (λ_em = 573.8 nm), while the reference b-CDs signal (λ_em = 466 nm) remained unaffected, confirming the ratiometric nature of the sensor. This linear behavior, comparable to that observed in PBS, suggests that the dual@nanoMIPs can function effectively even in diluted serum, although the use of undiluted or more complex fluids may still pose a challenge due to increased nonspecific interactions or optical interference. Notably, a more pronounced quenching effect was observed with dual@nanoMIPs imprinted at 10 kU mL⁻¹ CA19-9 compared to those imprinted at 1 kU mL⁻¹, demonstrating enhanced sensitivity (Figure S3, Table S2). The calculated LOD in 1% serum for the 10 kU mL⁻¹-imprinted dual@nanoMIPs was 2.40 × 10⁻³ U mL⁻¹ (S/N = 3), with a k_SV of –0.0942 and an imprinting factor (IF) of 3.01. The sensor exhibited a linear response from 4.98 × 10⁻³ U mL⁻¹ to 8.39 × 10² U mL⁻¹, encompassing the clinical diagnostic threshold for pancreatic cancer (CA19-9 > 37 U mL⁻¹) [48]. By contrast, dual@nanodots (non-imprinted) showed lower sensitivity and a LOD of 3.97 × 10⁻³ U mL⁻¹ with a smaller k_SV of –0.034, likely due to nonspecific adsorption of target molecules onto the negatively charged surface of y-QDs. While fluorescence quenching in controls indicates potential matrix-related interferences, the significantly higher response of imprinted materials demonstrates that the dual@nanoMIPs can effectively differentiate specific binding events from nonspecific interactions, even in a protein-rich environment like human serum. The matrix effect, although reduced through sample dilution, is further mitigated by the inclusion of an internal fluorescence reference (b-CDs), which compensates for fluctuations in signal intensity due to environmental variations such as pH, ionic strength, or sample turbidity. This ratiometric design contributes to improved sensor accuracy and reliability in complex biological fluids. A distinct fluorescence color gradient, observable under UV light in 1% human serum, reinforced the sensor’s potential for semi-quantitative visual detection. For future clinical applications, further optimization will be required for direct use in undiluted or minimally processed biological fluids, along with large-scale validation in patient-derived samples.
3、The mechanism of fluorescence quenching is proposed to involve charge transfer rather than Förster resonance energy transfer (FRET). However, no direct evidence (e.g., time-resolved fluorescence or computational modeling) is provided.The author should further explain the mechanism.
We thank the reviewer for highlighting the need to further clarify the proposed fluorescence quenching mechanism. As correctly pointed out, Förster resonance energy transfer (FRET) is unlikely due to the lack of spectral overlap between the emission of the quantum dots (QDs) and the absorption of CA19-9. In the revised version of the manuscript, we have expanded Section 3.1, as follows:
3.1. Assembly conditions of dual@nanodots and of dual@nanoMIPs
Steady-state fluorescence spectroscopy was employed to monitor the formation of dual@nanodots and to conduct subsequent calibration analyses, as illustrated in Figure 2. The yellow quantum dots (y-QDs) used in this study were surface-modified with 3-mercaptopropionic acid, which possesses two pKa values - 4.34 for the carboxylic acid group and 10.84 for the thiol, indicating that their stability may be compromised outside this pH range, likely due to oxidation. For the synthesis of blue-emitting carbon dots (b-CDs), the precursor solution was acidified to pH 4.0 prior to hydrothermal treatment. Mildly acidic environments are known to promote nitrogen doping and enhance fluorescence by supporting the protonation of amine groups and facilitating controlled carbonization of the citric acid–ethylenediamine matrix. These conditions favor better quantum yield and a narrow size distribution [40,44]. The emission spectrum of the assembled dual@nanodots in PBS buffer (10 mM, pH 7.4) displayed a primary emission peak at 574.9 nm corresponding to the y-QDs, while the b-CDs showed a less intense peak at 466 nm (Figure 2, blue line). Notably, fluorescence quenching was more significant in MIPs imprinted with a higher CA19-9 concentration (10 kU mL⁻¹, Figure 2d) than with a lower concentration (1 kU mL⁻¹, Figure 2b). Since CA19-9’s absorption spectrum (Figure S1, pink line) does not overlap with the emission wavelengths of either y-QDs or b-CDs, fluorescence quenching via Förster resonance energy transfer (FRET) is unlikely. Instead, the observed decrease in emission intensity, particularly for the yellow-emitting QDs, is more consistent with a photoinduced electron transfer (PET) or static quenching mechanism involving charge transfer. Specifically, upon excitation, electrons in the conduction band of the QDs may be transferred to electron-deficient moieties on the CA19-9 antigen, whose absorption falls within the same spectral region as the QDs' emission [42,43]. This nonradiative pathway likely contributes to the fluorescence suppression. The more pronounced quenching observed in MIPs imprinted at higher antigen concentrations (10 kU mL⁻¹) suggests a greater density of binding sites enabling close proximity between QDs and CA19-9, thereby enhancing electron transfer efficiency. Although time-resolved spectroscopy would offer more direct evidence of this mechanism, the current spectroscopic data and literature precedents provide strong support for a charge transfer-based quenching process.
4、The synthesis of b-CDs and y-QDs is briefly described, but key parameters (e.g., yield, reproducibility of quantum yield) are omitted.Please ask the author to supplement this part.
We appreciate the reviewer’s insightful comment regarding the insufficient detail on key synthetic parameters for the fluorescent nanoparticles. In response, we have thoroughly revised Sections 2.2, 2.3, 2.4, and 3.1 of the manuscript and we have introduced a new Section 2.2 titled Methods. In addition, we have complemented the explanation of the QY calculations in the Electronic Supplementary Material (ESM), as follows:
- Methods
UV-vis spectra were obtained in the interval range (430-730 nm) on an Evolution 220 UV-vis spectrophotometer (Thermo Scientific).
FTIR spectra were collected using a Nicolet iS10 spectrometer (Thermo Scientific) coupled to an attenuated total reflectance (ATR) sampling accessory of diamond contact crystal with a resolution of 16 cm-1 and a spectral range of (800 – 4000) cm-1.
Scanning Electron Microscopy (SEM) images of the conjugates were take non a FEI Quanta 400 FEG, operating at an accelerating voltage of 15 kV. EDS images were also obtained.
Fluorescence spectra were obtained on a Lumina fluorescence spectrometer (Thermo Scientific) equipped with a 150 W continuous wave xenon-arc discharge lamp as a light source at a scanning rate of 600 nm/min., a scan speed of 600 nm/min., an integration time of 50 ms and a response time of 0.02s. The photomultiplier tube voltage used was 310 PMT, both excitation and emission slits were set to 20 nm and both emission and excitation filters were set to air. The fluorescence measurements were performed with a Hellma suprasil quartz cell with a 1 mm optical path. Unless otherwise stated, fluorescence measurements of samples prepared from dual@nanodots were excited at 380 nm (λex = 380 nm) and its emissions registered between 430 nm to 730 nm.
- Synthesis of b-CDs
Blue-emitting carbon dots (b-CDs) were synthesized following a modified version of a previously published method [39,40]. Briefly, citric acid (1.1 g) and ethylenediamine (0.7 mL) were dissolved in water to achieve a 10% (w/v) solution. The pH was adjusted to 4.0 with 1 M HCl, and the solution was transferred into a 50 mL Teflon-lined stainless-steel autoclave for hydrothermal treatment at 260 °C for 4 hours (Figure 1A). After cooling to room temperature (22 °C), the resulting product was purified by dialysis (1000 Da cutoff membrane, Spectra/Por 6) against Milli-Q water for five days. The resulting b-CDs aqueous solution was stored in the dark at rt, as a stock solution and diluted in PBS to Abs(λₘₐₓ 320 nm) = 0.2 prior to further use.
- Synthesis of y-QDs
Yellow-emitting cadmium telluride (CdTe) quantum dots capped with 3-mercaptopropionic acid (MPA), referred to as y-QDs, were synthesized based on a modified procedure from Zhou et al. [41]. In a three-necked flask, CdCl₂ (4.2 × 10⁻³ mol) and MPA (7.2 × 10⁻³ mol) were added to 100 mL of ultrapure water, and the pH was adjusted to 11.5 using 1 M NaOH. A separate Te²⁻ precursor was prepared by reacting elemental tellurium (2.9 × 10⁻³ mol) with NaBH₄ (4.8 × 10⁻³ mol) in 3 mL of degassed ultrapure water at 100 °C. After 30 minutes of N₂ purging, the Te²⁻ solution was injected into the Cd²⁺/MPA solution. The optimal molar ratio Cd²⁺:Te²⁻:MPA was maintained at 1:0.1:1.7, and the reaction mixture was refluxed for 4 hours (Figure 1A). Ethanol precipitation and centrifugation at 4000 rpm for 5 minutes yielded the solid product, which was dried in the dark and stored.
- Assembly conditions of dual@nanodots and of dual@nanoMIPs
Steady-state fluorescence spectroscopy was employed to monitor the formation of dual@nanodots and to conduct subsequent calibration analyses, as illustrated in Figure 2. The y-QDs used in this study were surface-modified with 3-mercaptopropionic acid, which possesses two pKa values - 4.34 for the carboxylic acid group and 10.84 for the thiol, indicating that their stability may be compromised outside this pH range, likely due to oxidation. For the synthesis of b-CDs, the precursor solution was acidified to pH 4.0 prior to hydrothermal treatment. Mildly acidic environments are known to promote nitrogen doping and enhance fluorescence by supporting the protonation of amine groups and facilitating controlled carbonization of the citric acid–ethylenediamine matrix. These conditions favour better quantum yield and a narrow size distribution [40,44]. The resulting b-CDs aqueous solution had an average synthesis yield of 71.3 ± 3.1% (w/w, n = 3). Quantum yield (QY) was measured using quinine sulfate as a reference (QY = 54%) and was calculated to be 22.6 ± 1.4% in PBS (10 mM, pH 7.4, n = 3), as shown in ESM, Table S1. Reproducibility between batches was confirmed over three independent syntheses, yielding consistent optical properties (±6.2% variation in peak intensity and ±2.1 nm variation in emission maximum at 466 nm). The synthesis yield for y-QDs averaged 63.8 ± 2.6% (w/w, n = 3). The QY was estimated at 31.2 ± 2.3% using rhodamine 6G (QY = 95%) as a standard, as shown in ESM, Table S1. Inter-batch reproducibility showed less than 7.5% variation in intensity and a consistent emission maximum at 574.9 ± 1.9 nm. The emission spectrum of the assembled dual@nanodots in PBS buffer (10 mM, pH 7.4) displayed a primary emission peak at 574.9 nm corresponding to the y-QDs, while the b-CDs showed a less intense peak at 466 nm (Figure 2, blue line). Notably, fluorescence quenching was more significant in MIPs imprinted with a higher CA19-9 concentration (10 kU mL⁻¹, Figure 2d) than with a lower concentration (1 kU mL⁻¹, Figure 2b). Since CA19-9’s absorption spectrum (Figure S1, pink line) does not overlap with the emission wavelengths of either y-QDs or b-CDs, fluorescence quenching via Förster resonance energy transfer (FRET) is unlikely. Instead, the observed decrease in emission intensity, particularly for the yellow-emitting QDs, is more consistent with a photoinduced electron transfer (PET) or static quenching mechanism involving charge transfer. Specifically, upon excitation, electrons in the conduction band of the QDs may be transferred to electron-deficient moieties on the CA19-9 antigen, whose absorption falls within the same spectral region as the QDs' emission [42,43]. This nonradiative pathway likely contributes to the fluorescence suppression. The more pronounced quenching observed in MIPs imprinted at higher antigen concentrations (10 kU mL⁻¹) suggests a greater density of binding sites enabling close proximity between QDs and CA19-9, thereby enhancing electron transfer efficiency. Although time-resolved spectroscopy would offer more direct evidence of this mechanism, the current spectroscopic data and literature precedents provide strong support for a charge transfer-based quenching process. Regarding incubation time, a 1-hour exposure to CA19-9 was determined to be optimal, as extending beyond this period did not further alter the fluorescence response of the receptor-target imprinted polymers (rtiPs) (Figure 6a). A bathochromic shift in fluorescence emission was consistently observed post-polymerization (yellow lines) and following template removal (green lines) for both dual@nanoMIPs and dual@nanoNIPs (Figure 2b, d). Template removal was confirmed by fluorescence recovery of the dual@nanoMIPs after washing, as well as by the absence of CA19-9 absorbance in the UV/vis spectra of the wash supernatants (measured at 195 nm). The MIPs were washed using carbonate/bicarbonate buffer (10 mM, pH 9.8), and centrifugation-washing cycles were repeated until the fluorescence signal stabilized and CA19-9 was undetectable in the wash fractions. No UV irradiation was used during the removal process to avoid degradation of the fluorescent probes. However, only the imprinted materials showed partial recovery of fluorescence after washing, indicating successful antigen removal. This recovery was not observed in the non-imprinted controls (Figure 2a, c, green lines), reinforcing the specific nature of the imprinting. UV-Vis analysis of the wash solutions confirmed the absence of CA19-9 in the supernatants of dual@nanoMIPs (Figure S1, green dashed lines), suggesting effective template removal. The MIP synthesis employed a surface imprinting strategy to avoid deep embedding of the template and to facilitate accessible binding sites. This method minimizes the risk of total entrapment of CA19-9 within the polymeric matrix. To evaluate the effect of imprinting concentration on binding site formation, dual@nanoMIPs were prepared using two CA19-9 concentrations: 1 kU mL⁻¹ and 10 kU mL⁻¹. Fluorescence quenching was more pronounced in materials imprinted with the higher concentration (Figure 2d), supporting the hypothesis that greater imprinting densities enhance target rebinding. This may be explained by electrostatic interactions between the positively charged CA19-9 under physiological pH and the negatively charged surfaces of the y-QDs, given their known pKa values [45,46]. Conversely, fluorescence recovery after washing was again only observed for the imprinted polymers (Figure 2d green line), particularly following treatment with carbonate/bicarbonate buffer (10 mM, pH 9.8), confirming successful removal of the target molecule. Additional experimental details can be found in the ESM file.
Electronic Supplementary Material (ESM)
Quantum Yield (QY) calculations
The quantum yields (QYs) of the synthesized b-CDs and y-QDs were determined using a comparative method with well-established reference fluorophores. For b-CDs, quinine sulfate in 0.1 M H₂SO₄ () was used as the reference standard, while for y-QDs, rhodamine 6G in ethanol () was employed. All solutions were prepared to have absorbance values below 0.1 at the excitation wavelength (λex = 320 nm) for b-CDs and (λex = 390 nm) for y-QDs to minimize inner filter effects. All spectra were recorded at room temperature (22 °C). The absorbance of both sample and standard was kept below 0.1 at their corresponding excitation wavelengths to minimize inner filter effects. Solvent: Milli-Q water.
Fluorescence emission spectra were recorded under identical conditions, and the integrated emission intensities were used in the following equation:
Where:
is the quantum yield of the sample (b-CDs or y-QDs);
is the quantum yield of the reference fluorophore (standard):
For b-CDs: Quinine Sulfate in 0.1 M H₂SO₄ (Φ = 0.54)
For y-QDs: Rhodamine 6G in ethanol (Φ = 0.95)
and are integrated fluorescence intensities of the sample and reference
and are absorbances at the excitation wavelength (kept below 0.1 to avoid inner filter effects);
and are the refractive indices of solvents for sample and reference (usually the same if both in water or PBS), (both water, so ).
Experimental Conditions
All spectra were recorded at room temperature (22 °C)
Absorbance values were kept below 0.05 to minimize inner filter effects.
Solvents used: 0.1 M H₂SO₄ for quinine sulfate (n ≈ 1.33), ethanol for Rhodamine 6G (n ≈ 1.36), and PBS (n ≈ 1.33) for sample measurements.
All spectra were corrected for baseline and instrument response.
QY values and its standard deviation were averaged in triplicate, as shown in Table S1.
Results
The QY of b-CDs was calculated as 22.6 ± 1.4% using quinine sulfate as the reference.
The QY of y-QDs was calculated as 31.2 ± 2.3% using rhodamine 6G as the reference.
Table S1. Relative Quantum Yield Determination of b-CDs and y-QDs.
|
Sample |
Reference Fluorophore |
Φr |
λex (nm) |
Asample |
Aref |
Isample |
Iref |
Solvent (n) |
Φ (%) |
Std. Dev. (n = 5) |
|
b-CDs |
Quinine sulfate |
0.54 |
320 |
0.045 |
0.046 |
382.4 |
910.1 |
1.33 |
22.6 |
±1.4 |
|
y-QDs |
Rhodamine 6G |
0.95 |
390 |
0.042 |
0.044 |
679.2 |
2069.5 |
1.36 |
31.2 |
±2.3 |
5、The manuscript highlights potential applications in pancreatic cancer diagnostics but does not address clinical sample validation. How does the author consider this problem?
We appreciate the reviewer’s thoughtful observation regarding the lack of validation using clinical samples. We fully acknowledge this limitation and have addressed it explicitly in the revised Conclusions section, as highlighted in yellow in the revised manuscript and shown bellow for guidance. While the current study demonstrates the dual@nanoMIP sensor’s strong analytical performance, including high selectivity, broad dynamic range, and compatibility with 1% human serum in PBS, its application to real clinical samples remains an essential next step for translational relevance.
- Conclusions
This study presents the development of a dual-emission molecularly imprinted polymer (dual@nanoMIP) sensor designed for the selective detection of CA19-9, a clinically significant biomarker for pancreatic cancer. The sensor integrates blue-emitting carbon dots (b-CDs) as an internal reference and yellow-emitting quantum dots (y-QDs) as the responsive element, enabling ratiometric fluorescence detection to enhance signal accuracy and reliability in complex matrices. The dual@nanoMIPs demonstrated a broad dynamic detection range (4.98 × 10⁻³ to 8.39 × 10² U/mL), high selectivity, and compatibility with 1% human serum in PBS. These results suggest its potential for preliminary point-of-care screening and follow-up of patients at risk of pancreatic cancer recurrence. Although the current findings are promising, further validation using real clinical samples and comparative studies against established techniques such as ELISA are required. From an economic perspective, the addition of dual probes may increase production costs slightly compared to conventional MIPs, but the improved performance and diagnostic value could support clinical adoption. Future work will focus on clinical validation, integration into diagnostic platforms, and assessment of long-term stability for potential commercialization.

Round 2
Reviewer 1 Report
Comments and Suggestions for Authors
Dear Authors,
Thank you for submitting your manuscript titled “Dual-Emitting Molecularly Imprinted Nanopolymers for the Detection of CA19-9” to Biomedicines. Your work presents a timely and innovative approach to CA19-9 detection through the use of dual-emitting MIPs, integrating quantum dots and carbon dots in a ratiometric fluorescence format.
The sensor design is conceptually strong, and the broad detection range in serum is a valuable contribution to point-of-care diagnostics. However, to enhance the clarity, reproducibility, and scientific rigor of your manuscript, I recommend that the following revisions be addressed:
in introduction part:
-
The novelty of this study is not adequately emphasized until late in the introduction.
-
The rationale for combining MIPs with dual-emitting nanodots could be articulated more sharply.
-
Language issues hinder comprehension in places.
- Recommendation : to Improve narrative structure and highlight the knowledge gap and novelty early.
- Materials and Methods:
-
The organization of the methods is at times confusing (e.g., the sequencing of sections 2.3, 2.4, 2.5).
-
Important experimental conditions (e.g., concentrations, excitation wavelengths) are repeated unnecessarily.
-
Statistical methods are not clearly described
- Recommendation: Clarify structure and include brief notes on validation techniques (e.g., standard deviation, RSD interpretation).
- Results and Discussion
-
The discussion is overly descriptive in parts and lacks deeper mechanistic interpretation.
-
Some results (e.g., PET mechanism for quenching) are speculative without direct evidence (e.g., no time-resolved studies).
-
Figures are dense and should be restructured for clarity.
- Recommendation: Separate results and discussion for clarity; reduce redundancy and improve figure readability.
- Conclusion
-
Lacks a concise summary of achievements and future directions.
-
Recommendation: Add a more concise and definitive conclusion that highlights limitations and future work.
- Figure 2 & 6: Too dense; reduce number of panels per figure or add subfigures to improve visual clarity
If these revisions are carefully addressed, the manuscript has the potential to make a valuable contribution to the field of biosensing and clinical diagnostics.
Thank you again for the opportunity to review your work.
Author Response
Reply from Authors to Reviewer 1_ROUND 2
Journal: Biomedicines (ISSN 2227-9059)
Manuscript ID: biomedicines-3649176
Type: Article
Title: Dual-Emitting Molecularly Imprinted Nanopolymers for the Detection of CA19-9
Authors: Eduarda Rodrigues , Ana Xu , Rafael C. Castro , David S. M. Ribeiro , João Santos , Ana Margarida L. Piloto *
Section: Biomedical Engineering and Materials
Special Issue: Application of Biomedical Materials in Cancer Therapy
We appreciate the reviewer’s insightful suggestions. In response, all reviewer comments have been included in blue, while the authors’ responses are provided in black. Corresponding revisions in the manuscript have been incorporated and are highlighted in yellow for ease of reference.
Reviewer 1
Comments and Suggestions for Authors
Dear Authors,
Thank you for submitting your manuscript titled “Dual-Emitting Molecularly Imprinted Nanopolymers for the Detection of CA19-9” to Biomedicines. Your work presents a timely and innovative approach to CA19-9 detection through the use of dual-emitting MIPs, integrating quantum dots and carbon dots in a ratiometric fluorescence format.
The sensor design is conceptually strong, and the broad detection range in serum is a valuable contribution to point-of-care diagnostics. However, to enhance the clarity, reproducibility, and scientific rigor of your manuscript, I recommend that the following revisions be addressed:
in introduction part:
- The novelty of this study is not adequately emphasized until late in the introduction.
- The rationale for combining MIPs with dual-emitting nanodots could be articulated more sharply.
- Language issues hinder comprehension in places.
Recommendation:
- to Improve narrative structure and highlight the knowledge gap and novelty early.
Response to Reviewer 1 – Introduction Section
We thank Reviewer 1 for their thoughtful and constructive comments, particularly regarding the clarity and structure of the Introduction.
Response:
We agree and have revised the Introduction as follows:
The rapid and reliable detection of biological targets such as DNA, antibodies, and proteins is critical in modern biomedical research and clinical diagnostics, enabling timely assessment of health conditions and disease progression [1–6]. Among these biomarkers, carbohydrate antigen 19-9 (CA19-9) is particularly important due to its association with pancreatic cancer, a disease with high mortality rates primarily due to late-stage diagnosis. Although CA19-9 is not exclusively specific to pancreatic cancer, it is routinely used in clinical practice for early diagnosis and risk assessment, complementing imaging and other diagnostic modalities [7–9]. Traditional detection methods for CA19-9, including enzyme-linked immunosorbent assays (ELISA) [10,11], surface-enhanced Raman scattering (SERS) [12], fluorescence-based assays [13,14], electrochemical sensing [15–18], and mass-spectrometry [19,20], offer high sensitivity and specificity. However, these techniques typically require sophisticated instrumentation, complex sample preparation, and trained personnel, factors that limit their application in rapid screening and point-of-care (PoC) settings. In response, point-of-care tests (PoCTs) have gained attention for their potential to provide accessible, rapid, and user-friendly diagnostic solutions [21–24]. A central challenge in developing effective PoCTs lies in achieving high sensitivity and specificity using stable, low-cost biorecognition elements. Molecularly imprinted polymers (MIPs) have emerged as promising synthetic alternatives to antibodies in sensor design due to their excellent chemical stability, reusability, and tunable selectivity [25,26]. Despite these advantages, conventional MIP-based fluorescent sensors can suffer from signal instability and environmental interference, particularly when applied in complex biological matrices [27–29]. To address these limitations, this study presents a novel dual-emitting molecularly imprinted nanoparticle (dual@nanoMIPs) platform for the sensitive and selective detection of CA19-9. To the best of our knowledge, this is the first report integrating both blue-emitting carbon dots (b-CDs) and yellow-emitting quantum dots (y-QDs) into a MIP framework for ratiometric fluorescence sensing of CA19-9 [30]. The dual-emission design enables internal referencing, enhancing signal reliability by compensating for environmental fluctuations such as pH changes, temperature variations, and photobleaching [31–34]. The rationale behind this approach is twofold: first, the molecular specificity imparted by MIPs ensures targeted binding of CA19-9, while the integration of fluorescent nanodots enhances signal generation and readout accuracy [35–39]. Second, the use of ratiometric fluorescence—based on the intensity ratio between y-QDs (signal channel) and b-CDs (reference channel)—mitigates external interference and improves reproducibility compared to single-emission systems. While traditional QD-based immunoassays offer high brightness and tunable optical properties [40–42], their reliance on fragile and expensive biological recognition elements limits scalability and robustness. In contrast, MIP-based platforms offer a synthetic, cost-effective alternative, although they typically face challenges such as low imprinting efficiency or signal interference [25,26,43]. By combining the optical advantages of QDs with the robustness of MIPs and the stability of internal referencing via b-CDs, the proposed dual@nanoMIPs sensor overcomes these drawbacks. The resulting sensor exhibits high selectivity, minimal cross-reactivity, and a wide dynamic detection range (4.98 × 10⁻³ to 8.39 × 10² U mL⁻¹) in 1% human serum in PBS. This system represents a significant step forward in the development of next-generation PoC diagnostic tools for early pancreatic cancer screening, recurrence monitoring, and clinical decision-making.
Materials and Methods:
- The organization of the methods is at times confusing (e.g., the sequencing of sections 2.3, 2.4, 2.5).
- Important experimental conditions (e.g., concentrations, excitation wavelengths) are repeated unnecessarily.
- Statistical methods are not clearly described
Recommendation:
- Clarify structure and include brief notes on validation techniques (e.g., standard deviation, RSD interpretation).
Response:
We appreciate the suggestions to improve the clarity, organization, and scientific rigor of this section. In response, we have made the requested revisions, as highlighted in yellow in the revised manuscript.
- Materials and Methods
2.1. Materials
Tellurium powder (200 mesh, 99.8%), sodium borohydride (NaBH₄, 99%), cadmium chloride hemi(pentahydrate) (CdCl₂·2.5H₂O, 99%), sodium hydrogen carbonate (NaHCO₃), and sodium carbonate decahydrate (Na₂CO₃·10H₂O) were purchased from Sigma-Aldrich. Unconjugated human carbohydrate antigen CA19-9 (produced in E. coli, ~210 kDa) were obtained from Biorbyt Ltd., UK. Acrylamide (AAM), bisacrylamide (MBA), 2-aminoethyl methacrylate hydrochloride (AEMH·HCl), 3-mercaptopropionic acid (MPA, 99%), and tetramethyl ethylenediamine (TEMED) were purchased from TCI. Phosphate-buffered saline (PBS) was obtained from Amresco, ammonium persulfate (APS) from Analar Normapur, and human serum (HN, normal) from PZ CORMAY S.A., Poland. Human serum albumin (HSA), creatinine (CREA), matrix metalloproteinase 7 (MMP7), and carcinoembryonic antigen (CEA) were purchased from Sigma or Abbexa as appropriate. All solutions were prepared with Milli-Q water (specific conductivity <0.1 μS cm⁻¹). Chemicals were of analytical reagent grade and used without further purification. Fluorescence measurements employed a Hellma suprasil® quartz cell with a 1 mm optical path. Instrumentation details are provided in the Electronic Supplementary Material (ESM).
2.2. Methods
UV-Vis absorption spectra were recorded over 430–730 nm using a Thermo Scientific Evolution 220 spectrophotometer.
FTIR spectra were collected on a Nicolet iS10 spectrometer (Thermo Scientific) with ATR accessory (diamond crystal) at 16 cm⁻¹ resolution over 800–4000 cm⁻¹.
Scanning Electron Microscopy (SEM) and Energy Dispersive X-ray Spectroscopy (EDS) analyses were performed on a FEI Quanta 400 FEG at 15 kV.
Fluorescence spectra were recorded on a Lumina spectrometer (Thermo Scientific) with a 150 W xenon-arc lamp. Unless otherwise noted, samples were excited at 380 nm with emission recorded between 430–730 nm. Both excitation and emission slits were set to 20 nm, and measurements were conducted in 1 mm quartz cells.
2.3. Synthesis of Blue-Emitting Carbon Dots (b-CDs)
Blue-emitting carbon dots were synthesized following a modified hydrothermal method [39,40]. Citric acid (1.1 g) and ethylenediamine (0.7 mL) were dissolved in water to form a 10% (w/v) solution. The pH was adjusted to 4.0 using 1 M HCl. The solution was sealed in a 50 mL Teflon-lined autoclave and heated at 260 °C for 4 hours. After cooling to room temperature (~22 °C), the product was purified by dialysis (1000 Da cutoff) against Milli-Q water for five days. The purified b-CD solution was stored in the dark at room temperature and diluted in PBS to an absorbance of 0.2 at 320 nm prior to use.
2.4. Synthesis of Yellow-Emitting CdTe Quantum Dots (y-QDs)
Yellow-emitting CdTe quantum dots capped with MPA were prepared based on Zhou et al. [41] with slight modifications. CdCl₂ (4.2 × 10⁻³ mol) and MPA (7.2 × 10⁻³ mol) were dissolved in 100 mL ultrapure water, and the pH was adjusted to 11.5 with 1 M NaOH. Separately, a Te²⁻ precursor was generated by reacting tellurium powder (2.9 × 10⁻³ mol) with NaBH₄ (4.8 × 10⁻³ mol) in 3 mL degassed water at 100 °C under N₂ purging for 30 minutes. The Te²⁻ solution was rapidly injected into the Cd²⁺/MPA solution maintaining a molar ratio Cd²⁺:Te²⁻:MPA of 1:0.1:1.7, and the mixture was refluxed for 4 hours. The colloidal solution was precipitated with ethanol and centrifuged at 4000 rpm for 5 minutes at 22 °C. The collected solid was dried under dark conditions and stored away from light until use.
2.5. Preparation of Dual-Emitting Nanodots (dual@nanodots)
A suspension of y-QDs (1 mg mL⁻¹ in PBS 10 mM, pH 7.4) was mixed with 10 µL of the b-CD suspension (Abs at 320 nm = 0.2), and the final volume was adjusted to 1 mL with PBS. This mixture served as the dual-emission fluorescent probe for further polymer imprinting.
2.6. Synthesis of dual@nanoMIPs
Dual fluorescent molecularly imprinted nanopolymers (dual@nanoMIPs) were synthesized via a free-radical polymerization method under nitrogen to minimize oxidative inhibition. All aqueous solutions were deoxygenated by N₂ purging prior to use. To form the pre-polymerization complex, CA19-9 (1 kU mL⁻¹) was incubated with N-(3-aminopropyl)methacrylamide hydrochloride (AEMH·HCl, 33.7 mg mL⁻¹) in phosphate-buffered saline (PBS, 10 mM, pH 7.4) for 20 min at rt, facilitating non-covalent interactions between the template and functional monomer. Subsequently, yellow-emitting quantum dots (y-QDs, 1 mg mL⁻¹) were added, and the volume adjusted to 1 mL. The mixture was centrifuged (4000 rpm, 2 min, 22 °C), and the supernatant discarded to concentrate the complex. The resulting pellet was resuspended in PBS containing blue-emitting carbon dots (b-CDs, 10 µL, diluted to Abs(λ_max = 320 nm) = 0.2), along with acrylamide (AAM, 44.1 mg mL⁻¹), N,N′-methylenebisacrylamide (MBA, 22.3 mg mL⁻¹), ammonium persulfate (APS, 13.6 mg mL⁻¹), and tetramethylethylenediamine (TEMED, 13.3 mg mL⁻¹). The final volume was readjusted to 1 mL, and polymerization was carried out at rt for 30 min. After polymerization, nanoparticles were collected by centrifugation (4000 rpm, 2 min, 22 °C), and template removal was performed using carbonate/bicarbonate buffer (10 mM, pH 9.8). The particles were washed in three cycles of resuspension and centrifugation until UV–vis analysis confirmed the absence of free CA19-9 in the supernatant. For comparison, a second batch of dual@nanoMIPs was synthesized using a higher CA19-9 concentration (10 kU mL⁻¹) to evaluate imprinting efficiency and selectivity. Control non-imprinted polymers (dual@nanoNIPs) were prepared using the same protocol but without the template CA19-9 during the pre-complexation step. Final nanoparticles were stored as follows: for long-term storage, dried and kept protected from light at rt after water wash and centrifugation; for short-term use (≤15 days), stored in PBS (10 mM, pH 7.4) and protected from light at rt.
2.7. Statistical Analysis and Validation
All fluorescence measurements were conducted in triplicate unless otherwise stated. The relative standard deviation (RSD) was calculated to assess the precision and reproducibility of the results. Statistical significance was determined where applicable using a threshold of p < 0.05. Calibration curve fitting and determination of detection limits are presented in the Results section. To evaluate variability across concentration levels, a one-way ANOVA was performed, with statistical significance defined as p < 0.05. The corresponding RSD values and statistical outcomes are summarized in Table 2.
- Reproducibility and stability of the dual@nanoMIPs
Reproducibility was assessed using 1% human serum (HN) diluted in PBS. Four concentrations of CA19-9 standards (0.749, 4.31, 24.2, and 141 U mL⁻¹) were spiked into the serum solution, and 100 µL of dual@nanoMIPs suspension (1 mg mL⁻¹) was added to each. The total volume in each well was adjusted to 200 µL. After incubation, the suspensions were centrifuged (4000 rpm, 5 min, 22 °C), supernatants were discarded, and pellets were resuspended in fresh 200 µL serum solution.
To evaluate the stability of the dual@nanoMIPs imprinted with CA19-9 (10 kU mL⁻¹), fluorescence measurements were performed over time in PBS. A suspension of the dual@nanoMIPs (1 mg mL⁻¹) was prepared and 100 µL was dispensed into each well of a 48-well microplate. The final volume was brought to 200 µL with PBS, and fluorescence signals were recorded on days 0, 5, 10, and 20. Each measurement was carried out in triplicate (n = 3), and the relative standard deviation (RSD, %) was calculated to assess signal consistency. The same procedure was also applied to the corresponding non-imprinted controls dual@nanoNIPs.
Results and Discussion
- The discussion is overly descriptive in parts and lacks deeper mechanistic interpretation.
- Some results (e.g., PET mechanism for quenching) are speculative without direct evidence (e.g., no time-resolved studies).
- Figures are dense and should be restructured for clarity.
Recommendation:
- Separate results and discussion for clarity; reduce redundancy and improve figure readability.
Response:
We appreciate the reviewers’ constructive suggestions and have implemented the following revisions accordingly:
- The discussion is overly descriptive in parts and lacks deeper mechanistic interpretation.
- Some results (e.g., PET mechanism for quenching) are speculative without direct evidence (e.g., no time-resolved studies).
3.1. Assembly conditions of dual@nanodots and of dual@nanoMIPs
The assembly of dual@nanodots and their incorporation into dual@nanoMIPs was monitored via steady-state fluorescence spectroscopy. The y-QDs, surface-functionalized with 3-mercaptopropionic acid, exhibited an average emission maximum at 574.9 ± 1.9 nm with a quantum yield (QY) of 31.2 ± 2.3%, using rhodamine 6G as a reference standard (QY = 95%) [40,44]. The b-CDs, synthesized under mildly acidic conditions (pH 4.0), showed an emission peak at 466 nm and a QY of 22.6 ± 1.4%, using quinine sulfate as a standard (QY = 54%) [40,44]. Batch-to-batch reproducibility for both nanodots was consistent, with emission intensities varying by <7.5% and <6.2% for y-QDs and b-CDs, respectively (Table S1). In the assembled dual@nanodots system, both emissions were present in PBS (10 mM, pH 7.4), with y-QDs dominating the fluorescence profile. When these nanodots were embedded within the polymer matrix during MIP synthesis, fluorescence quenching was observed, most notably in MIPs imprinted with 10 kU mL⁻¹ of CA19-9 (Figure 2d). This effect was concentration-dependent, as lower imprinting (1 kU mL⁻¹) resulted in less quenching (Figure 2b). The quenching behavior was partially reversible upon template removal using carbonate/bicarbonate buffer (10 mM, pH 9.8), indicating successful elution of CA19-9 from the polymer matrix. Only imprinted polymers exhibited this recovery (Figure 2d, green line); no such change was observed in the non-imprinted controls (Figure 2c), underscoring the specificity of the binding. The significant fluorescence quenching in dual@nanoMIPs, particularly at higher CA19-9 imprinting concentrations, suggests the formation of specific binding sites that promote close proximity between the QDs and the target antigen. While the mechanism of quenching is not conclusively resolved in this study, it is unlikely to be Förster resonance energy transfer (FRET), given the lack of spectral overlap between CA19-9 absorbance and QD emission (Figure S1). Instead, the data are consistent with a static or photoinduced electron transfer (PET) process, whereby excited-state electrons from QDs may be transferred to electron-deficient groups on CA19-9. However, this interpretation remains speculative in the absence of time-resolved photophysical data, and future studies using fluorescence lifetime spectroscopy are needed to validate this mechanism [42,43]. The partial fluorescence recovery observed after template removal is attributed to the loss of specific quenching interactions following CA19-9 elution. The absence of this recovery in the non-imprinted controls supports the formation of true imprinted sites rather than nonspecific adsorption. The effectiveness of surface imprinting is further evidenced by the absence of residual CA19-9 in the supernatants, as confirmed by UV-Vis spectroscopy at 195 nm (Figure S1). The results suggest that imprinting concentration significantly influences site density and sensor performance. The more pronounced quenching in the 10 kU mL⁻¹ MIPs indicates improved rebinding due to increased site availability. Electrostatic interactions likely also contribute to this behavior, as CA19-9 is known to carry a net negative charge under physiological conditions, promoting association with the positively charged functional monomers and the negatively charged surfaces of the QDs [45,46]. Lastly, the 1-hour incubation time with CA19-9 was sufficient to reach equilibrium binding, as no further changes in fluorescence were observed with longer exposure (Figure 6a). This suggests that the dual@nanoMIPs exhibit rapid target recognition, a favorable property for diagnostic applications.
- Figures are dense and should be restructured for clarity.
3.7. Reproducibility and stability studies of dual@nanoMIPs
The dual@nanoMIPs demonstrated excellent analytical performance, with recovery values ranging from 99.86% to 107.25% and low RSDs between 0.84% and 1.90%, confirming high precision. A statistically significant difference (p < 0.05) between the measured and spiked value was observed only at the lowest concentration (0.749 U/mL), likely reflecting greater variability near the detection limit. For all other concentrations, p-values > 0.05 confirmed that the measured results were statistically indistinguishable from the true values, supporting the accuracy and reliability of the sensor.
Table 2. Recovery, RSD, and statistical significance of dual@nanoMIPs in 1% HN serum spiked with CA19-9.
|
Sample (U/mL) |
Spiked (U/mL) |
Mean Found (U/mL) |
RSD (%) |
Recovery (%) |
p-value |
|
1 |
0.749 |
0.803 |
1.90 |
107.25 |
0.0253 |
|
2 |
4.31 |
4.310 |
0.84 |
100.00 |
1.0000 |
|
3 |
24.2 |
24.167 |
1.67 |
99.86 |
0.8995 |
|
4 |
141.0 |
141.967 |
0.87 |
100.69 |
0.3078 |
In terms of stability, the dual@nanoMIPs retained 98.9 ± 2.1% of their initial fluorescence intensity after five days of storage in the dark, indicating excellent short-term photostability. Non-imprinted control particles (dual@nanoNIPs) retained 97.3 ± 2.3% over the same period, within acceptable limits (Figure 7b). These findings underscore the potential of the dual@nanoMIPs platform as a reproducible and stable sensing tool for CA19-9 detection in biological samples.
We thank the reviewer for the suggestion regarding figures. We would like to note that all Figures have already been revised and improved in response to similar comments from previous reviewers. These modifications included reducing panel density and enhancing visual clarity to ensure better readability. We trust the current versions meet the expectations for figure presentation, but we remain open to further suggestions if needed.
Conclusion
- Lacks a concise summary of achievements and future directions.
Recommendation:
- Add a more concise and definitive conclusion that highlights limitations and future work.
- Figure 2 & 6: Too dense; reduce number of panels per figure or add subfigures to improve visual clarity
If these revisions are carefully addressed, the manuscript has the potential to make a valuable contribution to the field of biosensing and clinical diagnostics.
Thank you again for the opportunity to review your work.
Response:
We appreciate the reviewer’s valuable feedback on the conclusion section and have revised it accordingly, as detailed below:
- Conclusions
This study reports the successful development of a dual-emission molecularly imprinted polymer sensor (dual@nanoMIPs) for selective and sensitive detection of CA19-9, an important biomarker for pancreatic cancer. By integrating blue-emitting carbon dots as an internal reference and yellow-emitting quantum dots as the responsive element, the sensor achieves reliable ratiometric fluorescence detection with a broad dynamic range (4.98 × 10⁻³ to 8.39 × 10² U/mL) and strong selectivity in 1% human serum. These findings demonstrate the sensor’s potential for early screening and monitoring of pancreatic cancer recurrence. However, further validation with clinical samples and comparison to gold-standard methods such as ELISA are necessary to confirm clinical utility. While the dual-probe design may marginally increase production costs, the enhanced diagnostic performance supports its translational promise. Future work will focus on clinical evaluation, integration into diagnostic platforms, and stability assessments to advance toward commercialization.
- Figure 2 & 6: Too dense; reduce number of panels per figure or add subfigures to improve visual clarity
We thank the reviewer for the suggestion regarding figures. We would like to note that Figures 2 and 6 have already been revised and improved in response to similar comments from previous reviewers. These modifications included reducing panel density and enhancing visual clarity to ensure better readability. We trust the current versions meet the expectations for figure presentation, but we remain open to further suggestions if needed.

Reviewer 2 Report
Comments and Suggestions for Authors
Thanks for your revisions.
Author Response
Thank you for your positive assessment and constructive feedback. We appreciate your recognition of the clarity in the English language, research design, methods, results presentation, conclusions, and figures. Your comments have been valuable in refining our manuscript, and we have carefully addressed all suggestions. We look forward to the continued evaluation process and the opportunity to contribute to the field.
Reviewer 3 Report
Comments and Suggestions for Authors
The author answered all my questions point-to-point and suggested to accept it.
Author Response
We appreciate the reviewer’s thorough evaluation and constructive feedback. Thank you for acknowledging our point-by-point responses. We have taken all your suggestions seriously and made the necessary improvements to enhance the manuscript. Your support towards acceptance is greatly appreciated.
Round 3
Reviewer 1 Report
Comments and Suggestions for Authors
Dear Authors,
Thank you for your thorough and thoughtful revision of the manuscript :Dual-Emitting Molecularly Imprinted Nanopolymers for the Detection of CA19-9.
I appreciate your clear and detailed responses to the comments I provided during the previous round of review. Your revisions to the abstract, introduction, methods, and conclusions effectively addressed the points I raised. Specifically, you clarified the intended clinical application of your platform, acknowledged limitations of MIPs, justified methodological parameters, and improved the organization and presentation of your experimental procedures.
The manuscript is now much improved in terms of both clarity and scientific rigor.
Best Regards